# Enteral Feeding Interventions in the Prevention of Necrotizing Enterocolitis: A Systematic Review of Experimental and Clinical Studies

**DOI:** 10.3390/nu13051726

**Published:** 2021-05-19

**Authors:** Ilse H. de Lange, Charlotte van Gorp, Laurens D. Eeftinck Schattenkerk, Wim G. van Gemert, Joep P. M. Derikx, Tim G. A. M. Wolfs

**Affiliations:** 1European Surgical Center Aachen/Maastricht, Department of Pediatric Surgery, School for Nutrition, Toxicology and Metabolism (NUTRIM), 6202 AZ Maastricht, The Netherlands; i.delange@maastrichtuniversity.nl (I.H.d.L.); wim.van.gemert@mumc.nl (W.G.v.G.); 2Department of Surgery, School for Nutrition, Toxicology and Metabolism (NUTRIM), Maastricht University, 6202 AZ Maastricht, The Netherlands; 3Department of Pediatrics, School of Oncology and Developmental Biology (GROW), Maastricht University, 6202 AZ Maastricht, The Netherlands; c.vangorp@maastrichtuniversity.nl; 4Department of Pediatric Surgery, Emma Children’s Hospital, Amsterdam UMC, University of Amsterdam and Vrije Universiteit Amsterdam, 1105 AZ Amsterdam, The Netherlands; l.eeftinckschattenkerk@amsterdamumc.nl (L.D.E.S.); j.derikx@amsterdamumc.nl (J.P.M.D.); 5Department of Biomedical Engineering (BMT), School for Cardiovascular Diseases (CARIM), Maastricht University, 6202 AZ Maastricht, The Netherlands

**Keywords:** necrotizing enterocolitis, enteral nutrition, inflammation, intestinal barrier function, microbial colonization

## Abstract

Necrotizing enterocolitis (NEC), which is characterized by severe intestinal inflammation and in advanced stages necrosis, is a gastrointestinal emergency in the neonate with high mortality and morbidity. Despite advancing medical care, effective prevention strategies remain sparse. Factors contributing to the complex pathogenesis of NEC include immaturity of the intestinal immune defense, barrier function, motility and local circulatory regulation and abnormal microbial colonization. Interestingly, enteral feeding is regarded as an important modifiable factor influencing NEC pathogenesis. Moreover, breast milk, which forms the currently most effective prevention strategy, contains many bioactive components that are known to support neonatal immune development and promote healthy gut colonization. This systematic review describes the effect of different enteral feeding interventions on the prevention of NEC incidence and severity and the effect on pathophysiological mechanisms of NEC, in both experimental NEC models and clinical NEC. Besides, pathophysiological mechanisms involved in human NEC development are briefly described to give context for the findings of altered pathophysiological mechanisms of NEC by enteral feeding interventions.

## 1. Introduction

Necrotizing enterocolitis (NEC) is a multifactorial disease, characterized by severe intestinal inflammation and, in advancing disease, gut necrosis, that mainly affects premature neonates [1]. Around 5 to 10% of very low birth weight (VLBW) infants develop NEC, with the highest incidence among neonates with an extremely low birth weight (ELBW) [2]. Despite advancing medical care, NEC incidence has not substantially decreased over time, mainly due to increased early survival of neonates [3,4,5]. NEC mortality is inversely correlated with birth weight and generally ranges from 15% to 30% [2,6]. However, case fatality can increase up to 50% for ELBW infants treated surgically [6,7]. Being responsible for 10% of NICU deaths, NEC represents an important cause of death in this setting [8]. Moreover, infants that do recover from NEC suffer from several long-term morbidities such as growth retardation [9], short bowel syndrome [10], intestinal failure [11], intestinal failure-associated liver disease and neurodevelopmental delays [12]. Although the precise healthcare costs of NEC are difficult to estimate [13], the costs undoubtedly exceed those of matched controls, with estimates of around $70,000 extra hospital costs for medical NEC and around $180,000 for surgical NEC [14]. Moreover, life-long care for patients with morbidities following NEC will impose an even higher financial burden on both society and the individual patient [15]. NEC thus forms an important health issue that has high impact on the patient and its parents and also leads to a significant economic burden.

Due to its complex pathophysiology and fulminant nature, NEC treatment remains, despite advancing medical care, largely symptomatic [1]. Moreover, effective prevention strategies are sparse [1]. Factors contributing to the excessive intestinal inflammation in NEC include immaturity of the intestinal immune defense, barrier function, motility and local circulatory regulation and abnormal microbial colonization [1,6]. Interestingly, NEC almost exclusively develops in infants that have been enterally fed and the NEC risk increases with delay of enteral feeding, indicating enteral feeding is an important target to modify NEC pathogenesis [16,17,18]. Breast milk contains many bioactive components that are known to shape neonatal (intestinal) immune development [19] and promote healthy gut colonization [20], thereby preventing intestinal inflammation [19]. Consequently, although not completely effective, breast milk is highly protective against NEC development and is currently considered the most effective preventive strategy [21,22]. Accordingly, several enteral feeding interventions that use donor breast milk or feeding components derived from breast milk have been studied over the past years as potential strategies for prevention of NEC [1,23]. This systematic review aims to describe the effect of different enteral feeding interventions on the prevention of NEC incidence and severity and the effect on pathophysiological mechanisms of NEC (intestinal inflammation, systemic inflammation, intestinal barrier function, vascular dysfunction/hypoxia-ischemia/free radical formation, intestinal epithelial cell death/altered proliferation, microbial dysbiosis, disturbed digestion and absorption and enteric nervous system alterations), in both experimental NEC models and clinical NEC. Besides, pathophysiological mechanisms involved in human NEC development are briefly described to contextualize the findings of altered pathophysiological mechanisms of NEC by enteral feeding interventions.

## 2. Materials and Methods

### 2.1. Search Strategy

To identify all relevant publications, the electronic databases PubMed, Embase and the Cochrane library were searched to select records published from inception until December 2020 that studied the effect of enteral feeding interventions in the prevention of NEC incidence and severity or pathophysiological mechanisms of NEC. An overview of the performed searches can be found in Appendix A. Both single and hierarchical search terms (e.g., MESH) were used. General search terms for nutritional interventions as well as terms for specific nutritional interventions often used in the context of NEC (expert opinion) (alkaline phosphatase (ALPI), epidermal growth factor (EGF)/heparin-binding EGF-like growth factor (HB-EGF), erythropoietin (EPO), exosomes, gangliosides, glutamine, immunoglobulins, insulin like growth factor (IGF), milk fat globule membrane, oligosaccharides, osteopontin, platelet-activating factor acetylhydrolase (PAF-AH), polyunsaturated fatty acids (PUFA), transforming growth factor β (TGF β), vitamin A and vitamin D) were incorporated in the search. For the search in the Cochrane library, results were filtered as to only retrieve Cochrane reviews. Last, references of included studies were cross-checked for additional studies that did not emerge in the original search. No restrictions were applied on study design or language. Results from the different searches were combined and after automatic removal of duplicates, the remaining records were screened for eligibility. No review protocol was published.

### 2.2. Selection Criteria

We included experimental animal studies (any experimental NEC model), RCTs and meta-analysis that reported on the effect of enteral feeding interventions on the prevention of NEC (incidence, severity (histological or clinical), NEC related mortality) or the prevention of pathophysiological mechanisms of NEC (intestinal inflammation, systemic inflammation, intestinal barrier function, vascular dysfunction/hypoxia-ischemia/free radical formation, intestinal epithelial cell death/altered proliferation, microbial dysbiosis, disturbed digestion and absorption and enteric nervous system alterations). Studies that did not relate to enteral feeding interventions as preventative treatment for NEC were excluded. Experimental studies with an enteral feeding intervention that started simultaneously with a NEC-inducing protocol were regarded as preventive. Studies that investigated intraperitoneal or intravenous administration were excluded (no enteral intervention). Regarding clinical studies, meta-analyses were included whenever possible. Meta-analyses of which a more recent or relevant (e.g., more studies included on NEC outcome) version was available, either by the same authors or different authors on the same subject, were excluded. RCTs were only included if: (1) a meta-analysis was not available or (2) the RCT was not included in a meta-analysis and was relatively large (*N* ≥ 50% of infants included in the meta-analysis) or (3) the RCT reported the effect of enteral feeding interventions on one of the pathophysiological mechanisms of NEC. RCTs that were excluded because of their low sample size relative to an earlier published meta-analysis are displayed in Appendix A [24,25,26,27]. Exclusion of these RCTs did not influence the findings and conclusions of this systematic review. No narrative reviews, in vitro studies, research protocols, comments on original articles, guidelines or conference abstracts were included.

### 2.3. Selection Process

Rayyan, online software enabling blind screening for reviewers [28], was used by to independent authors (I.H.d.L., C.v.G.) for article selection. Disagreements were solved by discussion. A third author was consulted in case consensus was not reached (T.G.A.M.W). In the first round, articles were screened based on title and abstract. In the second round, articles were full text screened.

### 2.4. Data Extraction

Data was extracted by one author (I.H.d.L.) from the included publications and corresponding Appendix A. When in doubt, inclusion of data was discussed with a second author (C.v.G.). All data related to the outcomes of interest (NEC incidence/severity/mortality and the pathophysiological hallmarks of NEC) were included. Data were first clustered based on type of study (experimental animal study or human trial), a second clustering was applied based on outcome reported and the last clustering was based on type of enteral feeding intervention (fat-based, carbohydrate/sugar-based, protein/amino acid-based, hormone/growth factor/vitamin-based, probiotic interventions and other interventions) (Figure 1). Additional parameters extracted were author, year of publication, experimental NEC model used (experimental animal studies), type of study (human studies), sample size, in- and exclusion criteria (human studies), intervention and control, sample size/power calculation and (primary and secondary) outcomes studied. For experimental animal studies, data are reported for the enteral feeding intervention group(s) compared to an untreated NEC protocol exposed group. For human studies, data are reported for the enteral feeding interventions treated group compared to an untreated (placebo) group.

### 2.5. Risk of Bias Assessment

The methodological quality and risk of bias of the different included studies were assessed with the use of the SYRCLE’s risk of bias tool [29] (experimental animal studies), the Jadad scoring system [30] (RCTs) and the AMSTAR measurement tool (meta-analyses) [31]. The assessment was performed by two independent authors (I.H.d.L. and L.D.E.S.). Disagreements were resolved by discussion.

### 2.6. Certainty of Evidence Assessment

Certainty of evidence of the effect of enteral feeding interventions tested in clinical studies (RCTs and meta-analyses) on NEC incidence or mortality was assessed with the GRADE approach [32]. These interventions were scored for limitations in study design or execution (risk of bias), inconsistency of results, indirectness of evidence, imprecision and the risk of publication bias by two independent authors (I.H.d.L. and C.v.G.). Disagreements were resolved by discussion. The scores on individual assessment points were combined in an overall estimation of certainty of evidence. Certainty of evidence is reported as “high” (we are very confident that the true effect lies close to that of the estimate of the effect), “moderate” (we are moderately confident in the effect estimate: the true effect is likely to be close to the estimate of the effect, but there is a possibility that it is substantially different), “low” (our confidence in the effect estimate is limited: the true effect may be substantially different from the estimate of the effect) or “very low” (we have very little confidence in the effect estimate: the true effect is likely to be substantially different from the estimate of effect) [32]. Certainty of evidence was not scored for animal studies. Although a GRADE scoring system for animal studies has been suggested [33], implementation of this methodology is still in its infancy and many aspects needed to adequately assess certainty of evidence from animal studies, such as 95% confidence intervals (CI) and power calculations, are seldomly reported.

## 3. Results

### 3.1. Study Characteristics

We identified a total number of 5883 records. After automatic removal of duplicates (1327 records), the remaining records (4573 records) were screened for eligibility (Figure 2). Of these articles, 4257 records were excluded in the first round. All of the remaining 316 articles could be retrieved. In the second round full-text screening, another 177 articles were excluded. An overview of the study characteristics of the included studies can be found in Appendix A (included experimental animal studies), Appendix A (included clinical trials), and Appendix A (included systematic reviews and meta-analyses). Whereas the risk of bias for the included animal studies (Appendix A) was in general unclear due to poor reporting of methodological details in these articles, the risk of bias for included RCTs (Appendix A) and meta-analyses (Appendix A) was predominantly low.

### 3.2. Enteral Feeding Interventions Affecting NEC Incidence and Severity in Animal Studies

Evidence of successful NEC prevention through enteral nutritional interventions in experimental animal models of NEC is abundantly present. In these models, many enteral nutritional interventions have been shown to reduce NEC incidence (Table 1), NEC severity (Table 2), clinical disease score or signs/symptoms (Table 3) and to improve survival (Table 4). Studies that did not report statistically significant preventative effects of enteral feeding interventions on NEC incidence, histological injury scores, clinical disease score or signs and symptoms or survival are summarized in Table 5. Importantly, supplementation of bovine lactoferrin increased the NEC severity score and elevated intestinal apoptosis and inflammation in a preterm pig NEC model [35,36], demonstrating that postulated beneficial enteral feeding interventions can also be harmful. This harmful effect may be caused by activation of the nuclear factor kappa-light-chain-enhancer of activated B cells (NFκβ) pathway and stimulation of interleukin 8 (IL8) release by enterocytes by a high dose of lactoferrin [35]. In addition, supplementation of formula with HB-EGF in a rat NEC model induced a dose dependent reduction of NEC incidence, with a therapeutic effect of moderate HB-EGF dosages that was not observed with either a low or a high HB-EGF dose [37]. This example highlights the importance of understanding the dose dependent working mechanisms of protective breast milk components. Some studies already provide mechanistic insight in the potential working mechanisms involved. For instance, the preventive effect of HMO was abolished in the presence of an inhibitor of the endoplasmic reticulum (ER) stress chaperone protein disulfide isomerase (PDI), suggesting PDI function is necessary for enteral HMO induced reduction of NEC incidence [38]. The protective effects of *Lactobacillus rhamnosus* on NEC severity score are toll like receptor 9 (TLR9) signaling dependent, as protective effects disappeared in TLR9 knock-down animals [39]. In addition, the protective effects of enteral administration of amniotic fluid in a mouse NEC model were demonstrated to be largely dependent on EGFR signaling, as the preventative effects mostly disappeared in the presence of the EGFR inhibitor cetuximab and with the use of amniotic fluid that was depleted of EGF [40]. Besides the supplemented substance and dose, timing and duration of the intervention are important. Addition of HB-EGF to all feeds, four feeds or two feeds per day reduced NEC incidence in a rat NEC model, while this was not the case when HB-EGF was only added to one feed per day [41]. In contrast to enteral HMO administration that was started within 24 h after birth and was continued during the duration of the study, enteral HMO administration that was started after the first 24 h or only given in the first 24 h did not result in improved histological NEC scores in a rat NEC model [42]. Similarly, enteral administration of HB-EGF successfully reduced NEC incidence when administration started within 12 h after birth, but not when supplementation was only initiated at or after 24 h [41]. Another interesting finding is that maternal feeding of a diet enriched with docosahexaenoic acid (DHA) or eicosapentaenoic acid (EPA) during pregnancy reduced NEC incidence in the offspring in a mouse NEC model [43], indicating that the fetus can already be targeted prenatally with a nutritional intervention to prevent NEC.

### 3.3. NEC Pathophysiology: Intestinal and Systemic Inflammation

Both intestinal and systemic inflammation are essential hallmarks of NEC pathophysiology. Acute NEC is characterized by increased intestinal expression of various cytokines, such as interleukin 1α (IL1α) [138], IL1β [139], TNFα [139], IL6 [140] and IL10 [140], whereas TGF-β tissue expression is decreased [108]. Intestinal cytokine levels normalize after recovery from NEC [139]. NEC is characterized by an increased number of polymorphonuclear leukocytes [141], neutrophil extracellular trap activation and release [142], and an increased number of macrophages in the intestine [141]. In addition, mRNA levels of C-X-C motif chemokine 5 (CXCL5), a chemokine stimulating influx of neutrophils were elevated in intestinal samples from infants with NEC compared to controls [141]. Moreover, a reduced proportion of functional regulatory T cells (Treg) in the intestine of NEC patients was observed compared to age-matched controls that was accompanied by a pro-inflammatory cytokine expression profile characteristic of inhibited Treg development [143]. As the proportion of Treg was restored after NEC recovery, it is likely that the strong inflammatory response during NEC temporarily inhibits Treg development [143]. In addition, an increased frequency of a subset of Treg, namely C-C motif chemokine receptor 9 (CCR9)-positive interleukin 17 (IL-17) producing Treg with strongly impaired immunosuppressive capacities, was found in peripheral blood during NEC and the conversion of CCR9+ Treg into this IL-17 producing subset was promoted by IL-6 [144]. Interestingly, in mice, treatment with anti-interleukin 6 receptor antibodies ameliorated NEC mortality, severity and morbidity and restored the balance between Treg and Th17 producing cells in peripheral blood, indicating a role for both cell types in the pathogenesis of NEC [144]. Altered expression and/or signaling of pattern recognition receptors (PRRs) is clearly involved in the pathogenesis of NEC [145]. Firstly, the role of toll like receptor 4 (TLR4) and some other TLRs, have been studied intensively in NEC pathogenesis [146]. In small intestinal specimen from infants with NEC, an increased mRNA expression [147,148] and increased protein levels [148,149] of TLR4 were found. Protein levels of TLR9 were reduced in the intestine of infants with NEC [149]. TLR4 knockout mice [147] as well as mice with a non-functioning mutation in TLR4 [148] are protected against experimental NEC. Reduced intestinal mRNA levels of negative regulators of TLR4 signaling (single IL1 receptor-related protein (SIGIRR), Toll-interacting protein (TOLLIP) and A20) have been observed in NEC [150] and mutations causing a loss of function of SIGGR are associated with NEC [151]. However, a prospective multicenter cohort study failed to show an association between genetic variants of TLR4, toll like receptor 2 (TLR2), toll like receptor 5 (TLR5), TLR9 or IL1 receptor-associated kinase 1 (IRAK1) and NEC [152]. Moreover, we previously reported that Myeloid Differentiation factor 2 (MD-2) could not be detected in the intestine or immune cells of infants with NEC, suggesting impaired LPS signaling [153]. This was confirmed by another study observing reduced protein levels of MD-2 and also TLR4 in the intestine of two NEC patients compared to control tissue from stoma closure of these two patients [154]. Last, nucleotide-binding oligomerisation domain (NOD)-like receptors are likely to be involved in NEC pathogenesis [145]. Mutations in the NOD2 gene, leading to loss of function, have been associated with an increased risk of severe NEC requiring surgery [151,155].

In addition to intestinal inflammation, infants with NEC have higher blood levels of pro-inflammatory mediators PAF [156], tumor necrosis factor α (TNFα), interleukin 6 (IL6) [157,158] and IL8 [157,158] and the anti-inflammatory cytokine interleukin 10 (IL10) [158]. Moreover, blood levels of IL6 [158], IL8 [158,159] and interleukin 1β (IL1β) [159] as well interleukin 1 receptor antagonist (IL1ra) [159] and IL10 [159] are higher in severe NEC compared to mild or moderate NEC [159]. Higher blood levels of interleukin 2 (IL2) and TGF-β are associated with a decreased NEC risk [157].

### 3.4. Enteral Feeding and Intestinal Inflammation in Animal Models of NEC

Intestinal inflammation is, in preclinical studies, the most extensively studied pathophysiological mechanism of NEC and many enteral feeding interventions reduce intestinal inflammation in animal models of NEC (Table 6).

#### 3.4.1. Fat-Based Feeding Interventions

Fat-based feeding interventions, such as polyunsaturated fatty acids (PUFA, including DHA, EPA, arachidonic acid (AA) and egg phospholipids), branched chain fatty acids (BCFA), bovine milk fat globule membrane (MFGM) and milk polar lipids (MPL), are extensively studied in relation to intestinal inflammation.

Supplementation of enteral feeding with fish oil, rich in n-3 PUFA such as DHA and EPA, prevents an increase in intestinal PAF and leukotriene B4 in a mouse NEC model [82] and partially prevents a rise in intestinal IL6 and TNFα protein expression in a rat NEC model [83]. Enrichment of formula feeding with DHA and arachidonic acid (AA) in a rat NEC model reduced intestinal mRNA levels of the PAF synthesizing enzyme phospholipase A2-II (PLA2) and of the PAF receptor (PAFR) [45]. Supplementation of egg phospholipids, AA and DHA or DHA alone lowers intestinal PAFR gene expression [44]. Enteral supplementation with egg phospholipids decreased gut TLR4 and ileal TLR2 mRNA expression in rats [44]. Finally, AA and DHA, but not DHA alone, lowered intestinal TLR4 mRNA expression, suggesting AA is the responsible agent for the found effects [44]. Interestingly, a maternal feeding intervention in rats with a DHA or EPA enriched diet during pregnancy resulted in increased levels of both DHA and EPA in the fetal intestine and reduced small intestinal mRNA expression of NFκβ inhibitor α (IκBα), NFκβ inhibitor β (IκBβ) and peroxisome proliferator-activated receptor ϒ (PPARϒ) in the offspring [43], demonstrating that postnatal gut inflammation can already be targeted prenatally. Addition of BCFA to rat formula feeding increases intestinal IL10 mRNA levels more than threefold and also enhances IL10 protein levels [46]. A very low fat or reduced long chain triacylglycerol diet (considered pre-digested as its digestion is not dependent on intestinal lipases) reduces intestinal mRNA expression of IL1β and TNFα [85]. Enteral supplementation of bovine MFGM reduces ileal mRNA expression of IL1β, TNFα and IL6, as well as protein expression of TLR4 [48], whereas TLR9 signaling remained unaffected [48]. Enteral treatment with MPL, which are abundantly present in MFGM, increased intestinal IL10 protein expression, while decreasing intestinal TNFα, IL6 and IL8 protein expression and TLR4 immunoreactivity [84]. In addition, MPL inhibited NEC induced intestinal p65 and p50 expression [84]. Pomegranate seed oil, rich in unsaturated fatty acids such as conjugated linolenic acids and oleic acid, blocks an increase in ileal gene expression of IL6, IL8, IL12, interleukin 23 (IL23) and TNFα in neonatal rats during NEC [47]. Taken together, numerous fat-based feeding interventions possess immune modulatory activities, making them promising candidates for NEC prevention in a clinical setting.

#### 3.4.2. Carbohydrate or Sugar-Based Feeding Interventions

Secondly, interventions using carbohydrate/sugar based dietary interventions have been shown to be successful in reducing intestinal inflammation, either by downregulating pro-inflammatory cytokines or by upregulating anti-inflammatory mediators. In a murine NEC model, addition of the neutral HMO 2′-fucosyllactose (2′-FL) to formula feeding reduced intestinal gene expression of IL6, IL1β and TLR4 [91]. Enteral administration of the HMO 2′-FL, 6′-sialyllactose (6′-SL) or a combination of both reduced intestinal mRNA levels of TNFα (murine and pig model), IL1β (pig model) and TLR4 (murine and pig model), while this effect was not observed with enteral administration of lactose [92]. In other studies, addition of HMO to formula feeding reduced ileal mRNA levels of IL6 [49], IL8 [49], IL1β [49] and TLR4 [49] and ileal protein levels of IL6 [49] and IL8 [49,88]. In addition, HMO reduced intestinal protein levels of phosphorylated NFκβ, phosphorylated IκBα and TLR4 [49]. In a preterm pig model of NEC, enteral administration of a mixture of four HMO increased small intestinal mRNA expression of IL10, IL12, TGF-β and TLR4, whereas other cytokines and TLR such as IL8, IFNϒ, TNFα and TLR2 were not affected [134]. Enteral administration of sialylated HMO (containing 6′-SL, 3′-SL and DSLNT) reduced ileal mast cell counts and dipeptidylpeptidase I (DPPI) activity and concomitantly reduced ileal protein levels of IL6 and TNFα [93]. Enteral administration of GOS/FOS decreased terminal ileum IL1β and TNFα protein levels and the mRNA expression of several pro-inflammatory cytokines including IL6, IL1β and TNFα in a rat NEC model [136]. NEC protocol exposed rats that are orally treated with ganglioside D3 (GD3) had lower ileal protein levels of TNFα, IL6, C-C motif chemokine ligand 5 (CCL5) and L-selectin, combined with higher protein levels of anti-inflammatory mediators TIMP metallopeptidase inhibitor 1 (TIMP1), IL1ra and IL10 than animals that were not treated with GD3 [50]. Furthermore, in the same rat model, protein expression of the Treg marker forkhead box P3 (FoxP3) was upregulated by the GD3 treatment and more ileal Foxp3+ cells were observed in the GD3 supplemented group [50].

#### 3.4.3. Protein or Amino Acid-Based Feeding Interventions

Various interventions using proteins or amino acids, such as IAP, lactoferrin, N-acetylcysteine, arginine and glutamine, have been used as nutritional interventions to reduce intestinal inflammation.

In a study using a neonatal rat NEC model, enteral administration of IAP preserved endogenous ileal IAP mRNA expression and dose dependently decreased ileal TNFα mRNA expression [160]. In preterm pigs, enteral bovine lactoferrin administration reduced proximal intestinal IL1β, but not IL8, protein levels [35]. Terminal ileum mRNA expression levels of IL6 and TNFα were reduced by enteral feeding supplemented with lactoferrin in a murine NEC model [101]. Oral administration of N-acetylcysteine reduced intestinal mRNA levels of IL1β and TNFα [85]. Arginine supplementation reduced ileal IL6 and TNFα mRNA levels [98]. Glutamine supplementation decreased intestinal protein concentrations of TNFα [84,95], IL6 and IL8 and decreased TLR4, p65 and p50 immunoreactivity [84], while increasing intestinal IL-10 protein concentrations [84]. In addition, upon enteral glutamine supplementation, mRNA and protein expression of TLR2 and TLR4 were lowered in ileum and colon, but not jejunum, of NEC protocol exposed rats [97].

#### 3.4.4. Hormone, Growth Factor or Vitamin-Based Feeding Interventions

Growth factors and hormones form another group of nutritional interventions with promising results regarding the reduction of intestinal inflammation in experimental models of NEC. In a NEC rat model, enteral administration of EGR decreased intestinal mRNA expression of interleukin 18 (IL18), while increasing mRNA expression of IL10 and the IL10 transcription factor specificity protein 1 (Sp1) [53]. Recombinant EGF from soybean extract reduced intestinal mRNA levels of cyclooxygenase 2 (COX-2) upon orogastric administration in a rat NEC model [104]. Gastric gavage of HB-EGF in a murine NEC model reduced the number of pro-inflammatory M1 and increased the number of immune modulatory M2 macrophages in the intestine [57]. Oral administration of TGF-β1 in a neonatal rat NEC model increased SMAD family member 2 (Smad2) activation/phosphorylation, reduced the number of phosphorylated NFκβ positive intestinal epithelial cells and prevented a NEC induced decrease of the NFκβ regulator IκBα [64]. Oral administration of IGF1 in a rat NEC model reduced intestinal TLR4 and NFκβ mRNA expression and IL6 protein expression [65].

Vitamins such as vitamin A are often studied as nutritional interventions in the context of NEC. Intragastric vitamin A supplementation significantly lowered intestinal IL6 and TNFα levels, both on protein and mRNA level, compared to NEC only animals [111]. Enteral treatment with all-trans retinoic acid (ATRA), a vitamin A metabolite, reduced ileal mRNA expression of IL6 and IL17 in a murine NEC model [110]. In addition, an increase of Treg (Foxp3+CD4+ T cells) and a decrease of CD4+Th17 cells upon enteral ATRA treatment was observed with fluorescence-activated cell sorting of lamina propria CD4+ T cells [110]. In another murine NEC study, enteral ATRA decreased the ileal mRNA expression of pro-inflammatory cytokines IL1β and IL6 [109]. Moreover, ATRA supplementation prevented NEC induced loss of Treg (preserved Foxp3 mRNA expression) and induction of Th17 cells (reduced IL17 mRNA expression) in CD4+ T cells isolated from the intestinal lamina propria [109]. In a mouse model of NEC, vitamin D decreased intestinal protein and mRNA expression of IL6, IL1β and TNFα [112].

#### 3.4.5. Probiotic Feeding Interventions

Probiotics are also a widely studied group of nutritional interventions. In a mice NEC model, reduction of terminal ileum IL1β [69,70] and TNFα [70] mRNA and protein levels upon oral administration of *Lactobacillus reteuteri* DSM 17938 was found. In a rat NEC model, both *Lactobacillus reuteri* DSM 17938 and *Lactobacillus reuteri* ATCC PTA 4659 reduced intestinal mRNA expression of TLR1, TLR4, IL6, TNFα and NFκβ and protein expression of TNFα, IL1β, TLR4 and phosphorylated Iκβ, while increasing the mRNA expression of IL10 [68]. Moreover, *Lactobacillus reuteri* DSM 17938 inhibited mRNA expression of the TLR interaction proteins mitogen-activated protein kinase 8 interaction protein 3 and increased NFκβ inhibitor-β, while *Lactobacillus reuteri* ATCC PTA 4659 inhibited myelin and lymphocyte protein mRNA expression (also TLR interaction protein) [68]. Supplementing formula with *Lactobacillus reuteri* DSM 17938 reduced the percentage of activated effector CD4+ T cells in the intestine, increased the proportion of CD4+ Foxp3+ Treg and tolerogenic dendritic cells in the gut and reduced intestinal protein levels of the pro-inflammatory cytokines IL1β and IFNϒ [67]. All these effects were TLR2 dependent, as they did not occur in TLR2 −/− mice [67]. In another study, *Lactobacillus reuteri* DSM 17938 increased the percentage of Foxp3+ CD4+ Treg cells and Foxp3+ CD4+ CD8+ Treg cells in the terminal ileum of rats, while decreasing the percentage of Foxp3+ CD4+ CD8+ Treg cells in the mesenteric lymph nodes, indicating migration of Tregs from the lymph nodes to the intestine following treatment with this probiotic agent [132]. In a murine NEC model, *Lactobacillus reuteri* DSM 17938 normalized the frequency of CD4+ Foxp3+ Treg cells in both ileum and mesenteric lymph nodes [69]. As most of these Treg in ileum as well as in the mesenteric lymph nodes were Helios positive, the cells are likely to be of thymic origin [69]. In addition, enteral treatment with *Lactobacillus reuteri* DSM 17938 reduced the increase of activated effector/memory T cells (CD44+CD45RBlo) and transitional effector T cells (CD44+CD45Rbhi) in the ileum during NEC [69]. Interestingly, enteral administration of *Lactobacillus reuteri* biofilms on sucrose or maltose loaded microspheres, but not administration of unbound *Lactobacillus reuteri,* reduced small intestinal mRNA levels of IL6, IL1β, C-C motif chemokine ligand 2 (CCL2), C-X-C motif chemokine 1 (CXCL1) and IL10 in a rat NEC model [72]. Enteral *Lactobacillus rhamnosus* GG, both in a low and higher dosage, reduced TLR4 expression (mRNA) and increased SIGIRR (mRNA, protein) and A20 (mRNA) levels [161]. In addition, mediators of the TLR4 signaling pathway phosphorylated IKKβ and phosphorylated p65 were reduced on protein level concomitant with a reduced intestinal inflammation on mRNA level (Intercellular Adhesion Molecule 1 (ICAM-1), IL8, IL1β) and protein level (ICAM-1, IL1β) [161]. The strain *Bifidobacterium bifidum* OLB6378 normalized ileum IL6 levels in NEC rats [73]. Orogastric administration of *Bifidobacterium infantis* reduced mRNA expression of the PAF synthesizing enzyme phospholipase-A2 II (PLA2 II) [75]. Intragastric administration of Bifidobacterium microcapsules in a rat NEC model reduced ileal protein expression of TLR4, TLR2 and NFκβ p65 [114]. Enrichment of formula feeding with *Bifidobacterium adolescentis* decreased ileal mRNA expression of TLR4, while increasing the mRNA expression of the negative regulators of TLR signaling TOLLIP and SIGIRR [76]. In addition, enteral administration of *Bacteroides fragilis* strain ZY-312 decreased intestinal IL1ß protein expression in a rat NEC model [113]. Enteral administration of *Bifidobacterium breve* M-16V reduced ileal mRNA levels of TLR4, IL1β, IL6, TNFα and IL10 and increased the mRNA levels of TLR2 in a rat NEC model [117]. In addition, ileal protein levels of macrophage inflammatory protein 1 α (MIP1α) and IL1β were increased by this intervention [117]. In a rat NEC model, enteral administration of *Saccharomyces boulardii* reduced terminal ileum protein concentrations of IL1β, IL6 and TNFα and the mRNA expression of several pro-inflammatory cytokines including IFNβ and TNFα [136]. Last, oral supplementation of the TLR9 ligand GpG-DNA, reduced terminal ileum IL6 mRNA expression in a murine NEC model [39]. In accordance with the extensive evidence on the immunomodulatory effect of probiotics in animal models of NEC, probiotics are currently the most promising enteral feeding intervention for the prevention of NEC in clinical practice.

#### 3.4.6. Other Enteral Feeding Interventions

Finally, several other food components have been linked to immune modulatory effects within the context of NEC. Ginger intake by rats with NEC reduces intestinal protein concentrations of IL1β, IL6, TNFα and myeloperoxidase (MPO) [121]. Enteral administration of fennel seed extracts reduces intestinal protein concentrations of MPO, TNFα and IL6 [122]. Bovine milk exosomes administered through gavage normalized terminal ileum protein expression of MPO in NEC mice [118]. Both native and pasteurized exosomes from human breast milk were able to reduce distal ileum IL6 mRNA levels and MPO activity (MPO protein levels) in a mouse NEC model [119]. Addition of rat amniotic fluid to formula feeding reduced ileal mRNA expression of the chemokines C-X-C motif chemokine 2 (CXCL2), CXCL5, CCL2, CCL5 and the pro-inflammatory cytokine IFNγ in rats that developed NEC [63]. In a preterm pig NEC model, enteral treatment with amniotic fluid reduced the distal small intestinal mRNA expression of IFNγ, IL1α and TNFα and middle small intestinal mRNA expression of IL1α, TNFα, IL6 and IL8 compared to formula fed pigs that developed NEC [123]. Oral administration of curcumin dose dependently reduced intestinal protein levels of IL1β, IL6, IL1, TNFα and protein and mRNA levels of TLR4 while increasing protein and mRNA levels of SIRT1 and nuclear factor erythroid 2-relatedfactor 2 (NRF2) [124]. In a rat NEC model, addition of surfactant protein A to formula feeding reduced ileal IL1β, TNFα and TLR4 protein levels, but did not affect ileal IFNϒ concentrations [80]. Administration of human β-defensin-3 in a rat NEC model reduced ileal mRNA expression of TNFα, IL6 and IL10 [81]. Enteral berberine reduced ileal protein concentrations of TLR4, IL6 and IL10 and reduced mRNA levels of TLR4, NFκβ and TNFα [79]. Finally, enteral administration of astragaloside IV, a flavonoid from the plant Astragalus membranaceaus dose dependently decreased mRNA levels of TNFα, IL6, IL1β and NFκβ p65, decreased MPO protein levels and decreased the phosphorylation rate of NFκβ p65 and that of IκBα in the distal ileum of NEC protocol exposed rats [126].

**Table 6 nutrients-13-01726-t006:** Effect of enteral feeding interventions that reduce intestinal inflammation in experimental animal models of NEC.

Enteral Feeding Intervention	Effect on Intestinal Inflammation (Compared to NEC Protocol Exposure without Feeding Intervention)
**Fat-based interventions**	
Fish oil (n-3 PUFA)	Intestinal PAF (protein) ↓ [82] Intestinal leukotriene B4 (protein) ↓ [82] Intestinal IL6 (protein) ↓ [83] Intestinal TNFα (protein) ↓ [83]
AA + DHA	Duodenal, jejunal and ileal TLR 4 (mRNA) ↓ [44] Intestinal TLR2 (mRNA) = [44] Intestinal PLA2-II (mRNA) ↓ [45] Intestinal PLA2-II (mRNA) = [44] Ileal, colonic and intestinal PAFR (mRNA) ↓ [44,45]
DHA	Intestinal TLR4 (mRNA) = [44] Intestinal TLR2 (mRNA) = [44] Intestinal PLA2-II (mRNA) = [44] Ileum and colon PAFR (mRNA) ↓ [44]
DHA (maternal intervention)	Ileal DHA ↑ [43] Ileal EPA ↑ [43] Small intestinal iκbα (mRNA) ↓ [43] Small intestinal IκBβ (mRNA) ↓ [43] Small intestinal PPARϒ (mRNA) ↓ [43]
EPA (maternal intervention)	Ileal DHA ↑ [43] Ileal EPA ↑ [43] Small intestinal IκBα (mRNA) ↓ [43] Small intestinal IκBβ (mRNA) ↓ [43] Small intestinal PPARϒ (mRNA) ↓ [43]
Egg phospholipids	Intestinal TLR 4 (mRNA) ↓ [44] Ileal TLR2 (mRNA) ↓ [44] Intestinal PLA2 (mRNA) = [44] Ileal and colonic PAFR (mRNA) ↓ [44]
BCFA	Ileal IL10 (mRNA) ↑ [46] Ileal IL10 (protein) ↑ [46]
Pomegranate seed oil	Ileal IL6 (mRNA) ↓ [47] Ileal IL8 (mRNA) ↓ [47] Ileal IL12 (mRNA) ↓ [47] Ileal IL23 (mRNA) ↓ [47] Ileal TNFα (mRNA) ↓ [47]
Pre-digested fat (less long chain triacylglycerol, not dependent on intestinal lipases)	Intestinal IL1β (mRNA) ↓ [85] Intestinal TNFα (mRNA) ↓ [85]
Very low-fat diet	Intestinal IL1β (mRNA) ↓ [85] Intestinal TNFα (mRNA) ↓ [85]
MFGM	Ileal IL6 (mRNA) ↓ [48] Ileal IL1β (mRNA) ↓ [48] Ileal TNFα (mRNA) ↓ [48] Ileal TLR4 (protein) ↓ [48]
MPL	Intestinal IL10 (protein) ↑ [84] Intestinal TNFα (protein) ↓ [84] Intestinal IL6 (protein) ↓ [84] Intestinal IL8 (protein) ↓ [84] Intestinal TLR4 (protein) ↓ [84] Intestinal p65 (protein) ↓ [84] Intestinal p50 (protein) ↓ [84]
**Carbohydrate/sugar-based interventions**	
HMO	Ileal IL6 (mRNA) ↓ [49] Ileal IL8 (mRNA) ↓ [49] Ileal IL1β (mRNA) ↓ [49] Ileal TLR4 (mRNA) ↓ [49] Ileal IL6 (protein) ↓ [49] Ileal IL8 (protein) ↓ [49,88] Ileal phosphorylated NFκβ (protein) ↓ [49] Ileal phosphorylated IκBα (protein) ↓ [49] Ileal TLR4 (protein) ↓ [49]
Mixture of four HMO	Small intestinal IL10 (mRNA) ↑ [134] Small intestinal IL12 (mRNA) ↑ [134] Small intestinal TGF-β (mRNA) ↑ [134] Small intestinal TLR4 (mRNA) ↑ [134] Small intestinal IL8 (mRNA) = [134] Small intestinal IFNϒ (mRNA) = [134] Small intestinal TNFα (mRNA) = [134] Small intestinal TLR2 (mRNA) = [134]
2′-FL	Intestinal IL6 (mRNA) ↓ [91] (Small) intestinal IL1β (mRNA) ↓ [91] [92] Small intestinal TNFα (mRNA) ↓ [92] (Small) intestinal TLR4 (mRNA) ↓ [91] [92]
6′-SL	Small intestinal IL1β (mRNA) ↓ [92] Small intestinal TNFα (mRNA) ↓ [92] Small intestinal TLR4 (mRNA) ↓ [92]
2′-FL + 6′-SL	Small intestinal IL1β (mRNA) ↓ [92] Small intestinal TNFα (mRNA) ↓ [92] Small intestinal TLR4 (mRNA) ↓ [92]
Sialylated HMO	Ileal mast cell counts ↓ [93] Ileal DPPI activity ↓ [93] Ileal IL6 (protein) ↓ [93] Ileal TNFα (protein) ↓ [93]
GOS/FOS	Terminal ileum IL1β (protein) ↓ [136] Terminal ileum TNFα (protein) ↓ [136] Terminal ileum IL1β (mRNA) ↓ [136] Terminal ileum TNFα (mRNA) ↓ [136] Terminal ileum IL6 (mRNA) ↓ [136]
GD3	Ileal TNFα (protein) ↓ [50] Ileal IL6 (protein) ↓ [50] Ileal CCL5 (protein) ↓ [50] Ileal L-selectin (protein) ↓ [50] Ileal TIMP1 (protein) ↑ [50] Ileal IL1ra (protein) ↑ [50] Ileal IL10 (protein) ↑ [50] Ileal Foxp3 (protein) ↑ [50] Ileal Foxp3 cellcount ↑ [50]
**Protein/amino acid-based interventions**	
IAP	Ileal endogenous IAP (mRNA) ↑ [160] Ileal TNFα (mRNA) ↓ [160]
L-Glutamine/glutamine	Intestinal TNFα (protein) ↓ [95] Intestinal IL10 (protein) ↑ [84] Intestinal TNFα (protein) ↓ [84] Intestinal IL6 (protein) ↓ [84] Intestinal IL8 (protein) ↓ [84] Intestinal TLR4 (protein) ↓ [84] Intestinal p65 (protein) ↓ [84] Intestinal p50 (protein) ↓ [84] Jejununal, ileal and colonic TLR4 (protein) ↓ [97] Jejununal, ileal and colonic TLR4 (mRNA) ↓ [97] Jejununal, ileal and colonic TLR2 (protein) ↓ [97] Jejununal, ileal and colonic TLR2 (mRNA) ↓ [97]
Arginine	Ileal IL6 (mRNA) ↓ [98] Ileal TNFα (mRNA) ↓ [98]
N-Acetylcysteine	Intestinal IL1β (mRNA) ↓ [85] Intestinal TNFα (mRNA) ↓ [85]
Lactoferrin	Ileal IL6 (mRNA) ↓ [101] Ileal TNFα (mRNA) ↓ [101]
Bovine lactoferrin	Proximal small intestinal IL1β (protein) ↓ [35]
**Hormone/growth factor/vitamin based interventions**	
EGF	Ileal IL18 (mRNA) ↓ [53] Ileal IL10 (mRNA) ↑ [53] Ileal Sp1 (mRNA) ↑ [53]
Recombinant EGF from soybean extract	Ileal COX2 (mRNA) ↓ [104]
HB-EGF	Intestinal M1 macrophages cellcount (CD86) ↓ [57] Intestinal % M1 macrophages/total macrophages (CD86/CD68) ↓ [57] Intestinal M2 macrophages cellcount (CD206) ↑ [57] Intestinal % M1 macrophages/total macrophages (CD206/CD68) ↑ [57]
TGF-β1	Ileal Smad2 activation/phosphorylation ↑ [64] Ileal phosphorylated NFκβ positive intestinal epithelial cells ↓ [64] Ileal IκBα (protein) ↑ [64]
IGF1	Ileal TLR4 (mRNA) ↓ [65] Ileal NFκβ (mRNA) ↓ [65] Ileal IL6 (protein) ↓ [65]
Vitamin A	Intestinal IL6 (protein) ↓ [111] Intestinal TNFα (protein) ↓ [111]
ATRA	Foxp3 (mRNA) in CD4+ T cells from lamina propria ↑ [109] IL17 (mRNA) in CD4+ T cells from lamina propria ↓ [109] FoxP3+ CD4+ T cells from lamina propria (FACs) ↑ [110] CD4+ Th17 cells from lamina propria (FACs) ↓ [110] Ileal IL1β (mRNA) ↓ [109] Ileal IL6 (mRNA) ↓ [109,110] Ileal IL17 (mRNA) ↓ [110]
Vitamin D	Intestinal ÌL6 (mRNA) ↓ [112] Intestinal IL1β (mRNA) ↓ [112] Intestinal TNFα (mRNA) ↓ [112] Intestinal ÌL6 (protein) ↓ [112] Intestinal IL1β (protein) ↓ [112] Intestinal TNFα (protein) ↓ [112]
**Probiotic interventions**	
*Lactobacillus reuteri* DSM 17938	Intestinal % CD4+ Foxp3+ Treg ↑ [67,69,132] Mesenteric lymph nodes % CD4+ Foxp3+ Treg ↑ [69] Terminal ileum % Foxp3+ CD4+CD8+ Treg cells ↑ [132] Mesenteric lymph nodes % Foxp3+ CD4+CD8+ Treg cells ↓ [132] Intestinal % tolerogenic DC ↑ [67] Intestinal % activated CD4+ Teff ↓ [67] Intestinal % activated effector/memory T cells (CD44+CD45RBlo) ↓ [69] Intestinal % transitional effector T cells (CD44+CD45RBhi) ↓ [69] Ileal IL10 (mRNA) ↑ [68] Ileal IL6 (mRNA) ↓ [68] Ileal TNFα (mRNA) ↓ [68,70] Ileal TLR4 (mRNA) ↓ [68] Ileal TLR1 (mRNA) ↓ [68] Ileal NFκβ (mRNA) ↓ [68] Ileal IL1β (mRNA) ↓ [69,70] Ileal IL1β (protein) ↓ [67,68,69,70] Ileal IFNϒ (protein) ↓ [67] Ileal TNFα (protein) ↓ [68,70] Ileal TLR4 (protein) ↑[68] Ileal phosphorylated Iκβ (protein) ↑[68] Ileal mitogen-activated protein kinase 8 interaction protein 3 (mRNA) ↓ [68] Ileal NFκβ inhibitor-β (mRNA) ↑ [68]
*Lactobacillus reuteri* ATCC PTA 4659	Ileal IL6 (mRNA) ↓ [68] Ileal TNFα (mRNA) ↓ [68] Ileal TLR4 (mRNA) ↓ [68] Ileal TLR1 (mRNA) ↓ [68] Ileal NFκβ (mRNA) ↓ [68] Ileal TNFα (protein) ↓ [68] Ileal IL1β (protein) ↓ [68] Ileal TLR4 (protein) ↑ [68] Ileal phosphorylated Iκβ (protein) ↑ [68] Ileal IL10 (mRNA) ↑ [68] Ileal myelin and lymphocyte protein (mRNA) ↓ [68]
*Lactobacillus rhamnosus* GG	Ileal TRL4 (mRNA) ↓ [161] Ileal SIGIRR (mRNA) ↑ [161] Ileal SIGIRR (protein) ↑ [161] Ileal A20 (mRNA) ↑ [161] Ileal p-IKKb (protein) ↓ [161] Ileal p-p65 (protein) ↓ [161] Ileal ICAM-1 (protein) ↓ [161] Ileal ICAM-1 (mRNA) ↓ [161] Ileal IL1β (protein) ↓ [161] Ileal IL1β (mRNA) ↓ [161] Ileal IL8 (mRNA) ↓ [161]
*Lactobacillus reuteri* DSM 20016	Small intestinal IL6 (mRNA) = [72] small intestinal IL1β (mRNA) = [72] Small intestinal CCL2 (mRNA) = [72] Small intestinal CXCL1 (mRNA) = [72] Small intestinal IL10 (mRNA) = [72]
*Lactobacillus reuteri* biofilm on sucrose loaded microspheres	Small intestinal IL6 (mRNA) ↓ [72] small intestinal IL1β (mRNA) ↓ [72] Small intestinal CCL2 (mRNA) ↓ [72] Small intestinal CXCL1 (mRNA) ↓ [72] Small intestinal IL10 (mRNA) ↓ [72]
*Bifidobacterium bifidum* OLB6378	Ileal IL6 (mRNA) ↓ [73]
*Bifidobacterium infantis*	Intestinal PLA2 II (mRNA) ↓ [75]
Bifidobacterium microcapsules	Ileal TLR4 (protein) ↓ [114] Ileal TLR2 (protein) ↓ [114] Ileal NFκβ p65 (protein) ↓ [114]
*Bifidobacterium adolescentis*	Ileal TLR4 (mRNA) ↓ [76] Ileal TOLLIP (mRNA) ↑ [76] Ileal SIGIRR (mRNA) ↑ [76]
*Bifidobacterium breve* M-16V	Ileal TLR4 (mRNA) ↓ [117] Ileal IL1β (mRNA) ↓ [117] Ileal IL6 (mRNA) ↓[117] Ileal TNFα (mRNA) ↓ [117] Ileal IL10 (mRNA) ↓ [117] Ileal TLR2 (mRNA) ↑ [117] Ileal MIP1α (protein) ↓ [117] Ileal IL1β (protein) ↓ [117]
*Bacteroides fragilis* ZY-312	Intestinal IL1β (protein) ↓ [113]
*Saccharomyces Boulardii*	Terminal ileum IL1β (protein) ↓ [136] Terminal ileum IL6 (protein) ↓ [136] Terminal ileum TNFα (protein) ↓ [136] Terminal ileum IFNβ (mRNA) ↓ [136] Terminal ileum TNFα (mRNA) ↓ [136]
CpG-DNA	Ileal IL6 (mRNA) ↓ [39]
**Other interventions**	
Ginger	Intestinal IL1β (protein) ↓ [121] Intestinal IL6 (protein) ↓ [121] Intestinal TNFα (protein) ↓ [121] Intestinal MPO (protein) ↓ [121]
Fennel seed extracts	Intestinal IL6 (protein) ↓ [122] Intestinal TNFα (protein) ↓ [122] Intestinal MPO (protein) ↓ [122]
Bovine milk exosomes	Distal ileal MPO (protein) ↓ [118]
Human milk exosomes	Ileal IL6 (mRNA) ↓ [119] Ileal MPO (protein) ↓ [119]
Amniotic fluid	Ileal CXCL2 (mRNA) ↓ [63] Ileal CXCL5 (mRNA) ↓ [63] Ileal CCL2 (mRNA) ↓ [63] Ileal CCL5 (mRNA) ↓ [63] Ileal IFNγ (mRNA) ↓ [63] Distal small intestinal IFNγ (mRNA) ↓ [123] Distal small intestinal IL1α (mRNA) ↓ [123] Distal small intestinal TNFα (mRNA) ↓ [123] Middle small intestinal IL1α (mRNA) ↓ [123] Middle small intestinal TNFα (mRNA) ↓ [123] Middle small intestinal IL6 (mRNA) ↓ [123] Middle small intestinal IL8 (mRNA) ↓ [123]
Curcumin	Intestinal IL1β (protein) ↓ [124] Intestinal IL6 (protein) ↓ [124] Intestinal IL18 (protein) ↓ [124] Intestinal TNFα (protein) ↓ [124] Intestinal TLR4 (protein) ↓ [124] Intestinal SIRT1 (protein) ↑ [124] Intestinal NRF2 (protein) ↑ [124] Intestinal TLR4 (mRNA) ↓ [124] Intestinal SIRT1 (mRNA) ↑ [124] Intestinal NRF2 (mRNA) ↑ [124]
Surfactant protein A	Ileal IL1β (protein) ↓ [80] Ileal TNFα (protein) ↓ [80] Ileal IFNϒ (protein) ↓ [80] Ileal TLR4 (protein) ↓ [80]
Human β-defensin-3	Ileal TNFα (mRNA) ↓ [81] Ileal IL6 (mRNA) ↓ [81] Ileal IL10 (mRNA) ↓ [81]
Berberine	Ileal TLR4 (protein) ↓ [79] Ileal IL6 (protein) ↓ [79] Ileal IL10 (protein) ↓ [79] Ileal TLR4 (mRNA) ↓ [79] Ileal NFκβ (mRNA) ↓ [79] Ileal TNFα (mRNA) ↓ [79]
Astragaloside IV	Distal ileal TNFα (mRNA) ↓ [126] Distal ileal IL1β (mRNA) ↓ [126] Distal ileal IL6 (mRNA) ↓ [126] Distal ileal NFκβ p65 (mRNA) ↓ [126] Distal ileal MPO (protein) ↓ [126] Distal ileal p-NFκβ p65/ NFκβ p65 (protein) ↓ [126] Distal ileal p-IκBα/ IκBα (protein) ↓ [126] Distal ileal p-IκBα (protein) ↓ [126] Distal ileal p-NFκβ p65 (protein)↓ [126] Distal ileal NFκβ p65 (protein) ↓ [126] Distal ileal IκBα (protein) ↑ [126]

↑ depicts an increase, ↓ depicts a decrease; PUFA, polyunsaturated fatty acids; AA, arachidonic acid; DHA, docosahexaenoic acid; EPA, eicosapentaenoic acid; BCFA, branched chain fatty acids; MFGM, milk fat globule membrane; MPL, milk polar lipids; HMO, human milk oligosaccharides; 2′-FL, 2′-fucosyllactose; 6′-SL, 6′-sialyllactose; GOS, galacto-oligosaccharides; FOS, fructo-oligosaccharides; GD3, ganglioside D3; IAP, intestinal alkaline phosphatase; EGF, epidermal growth factor; HB-EGF, hemoglobin-binding EGF-like growth factor; HGF, hepatocyte growth factor; TGF-β1, transforming growth factor β1; IGF1, insulin-like growth factor 1; ATRA, all-*trans*-retinoic acid.

### 3.5. Enteral Feeding and Systemic Inflammation in Animal Models of NEC

Although not frequently reported in literature, several enteral nutritional interventions have been shown to (partially) prevent systemic inflammation in experimental models of NEC (Table 7). In a murine NEC model, enteral administration of HMO significantly reduced systemic IL8 levels [49,88], while this was not seen with infant formula oligosaccharides [49]. Oral administration of TGF-β in a rat NEC model reduced serum levels of IL6 and interferon γ (IFNϒ) [64]. In a rat NEC model, a dose dependent decrease of serum TNFα, IL1β, and IL6 was observed upon enteral IAP treatment [162]. Enrichment of enteral nutrition with hyaluronan 35 kD, a glycosaminoglycan present in human milk, reduced plasma concentrations of the pro-inflammatory cytokines TNFα, C-X-C motif chemokine 1 (CXCL1), interleukin 12 p70 (L12p70), IL6 and in the high dosage group also IFNϒ in a murine NEC model [94]. Oral pre-treatment with *Bacteroides fragilis* strain ZY-312 in a *Cronobacter*
*sakazakii* induced rat NEC model reduced serum concentrations of TNFα and IFNϒ and increased the levels of the anti-inflammatory cytokine IL10 [113]. Berberine administration reduced serum concentrations of IL6 and IL10 in a rat NEC model [79]. Human β-defensin-3 partially prevented an increase in systemic TNFα concentrations in a rat NEC model [81]. Last, administration of the flavonoid astragaloside IV decreased serum protein concentrations of TNFα, IL6 and IL1β in a rat NEC model [126].

### 3.6. NEC Pathophysiology: Loss of Intestinal Barrier Function

The intestinal barrier consists of several parts that together protect the host against luminal microbiota and their toxins, while preserving the capacity to absorb nutrients [163]. It is formed by a biofilm of commensal bacteria, a mucus barrier, antimicrobial peptides (AMPs) secreted by enterocytes and Paneth cells, secretory IgA released by plasma cells and intestinal epithelial cells that are interconnected by an apical junction complex containing adherence junctions, desmosomes and tight junctions (TJ) [163]. TJ regulate paracellular permeability and consist amongst others of claudins, occludin, junctional adhesion molecules (JAM) and zonulae occludens (ZO) proteins [163]. Importantly, regulation of paracellular permeability by TJ proteins is a complex process, in which some proteins reduce permeability (such as occludin) while others promote permeability (such as claudin-2) [164,165]. In premature infants, several components of the intestinal barrier are still immature predisposing them to NEC development [6,166]. During NEC, these components are further impaired, resulting in a defective barrier function. The mucus barrier is affected during NEC; in severely damaged regions of human NEC biopsies fewer goblet cells are present [167,168], whereas in mildly injured regions similar or even increased numbers of goblet cells are observed [167]. In addition, reduced numbers of Paneth cells have been described in human NEC [167,168] and increased mRNA expression of defensin A5 and A6, an unaffected protein expression of defensin A5 [169] and decreased protein expression of defensin A6 [170]. In fecal samples, the percentage of intestinal bacteria bound by IgA negatively correlates with NEC development [171]. In biopsies from infants with NEC, transepithelial electrical resistance was lower and flux of mannitol was higher, indicating increased intestinal permeability compared to controls [172]. Reported alterations of apical junction complex proteins in human NEC specimen include a reduced mRNA expression of occludin [172,173], claudin-4 [173], vinculin [173] and ZO-1 [173], reduced immunoreactivity of occludin and ZO-1 in jejunum and ileum [173], increased immunoreactivity for claudin-2 in both colon and small intestine [116] and an increased protein expression and internalization of claudin-2 [174]. Of note, one study did not find differences in expression or distribution of occludin and ZO-1 [116].

### 3.7. Enteral Feeding and Loss of Intestinal Barrier Function in Animal Models of NEC

Many enteral feeding interventions have been studied in the context of NEC induced intestinal barrier loss (Table 8). Often, both structural (such as TJ expression and goblet cell counts) and functional read-outs were studied.

#### 3.7.1. Fat-Based Feeding Interventions

PUFA is the only fat-based feeding intervention that has been studied in relation to intestinal barrier function in NEC. Enteral treatment with PUFA (AA and DHA) reduced endotoxemia, as a read-out for barrier function loss, after 48 h in a rat NEC model, an effect that was interestingly abolished by additional supplementation with nucleotides [45]. Enteral supplementation of DHA in a rat NEC model resulted in a less permeable mucus barrier, reflected by reduced effective diffusivity of amine and carboxyl modified particles, less linear movements of *Escherichia coli* through intestinal mucus and reduced Escherichia coli movement speed through intestinal mucus [131]. Mucus contained less sialic acid upon DHA administration, but mucus structure, analysed with confocal imaging and scanning electron microscopy (SEM), was hardly altered by DHA administration [131].

#### 3.7.2. Carbohydrate or Sugar-Based Feeding Interventions

Secondly, carbohydrate or sugar-based dietary interventions have been studied. In mice, hyaluronan 35 kD in both a low (15 mg/kg) and high (30 mg/kg) dose prevented NEC induced increase in gut permeability, measured with oral administration of fluorescein isothiocyanate (FITC)-labelled dextran 4 kD and in the higher dose also reduced bacteraemia [94]. In addition, hyaluronan 35 kD treatment increased the expression of the TJ proteins occludin, claudin-2, -3 and -4 and ZO-1 both in control and NEC protocol treated animals and the localization of occludin and claudin-3 were normalized in these animals [94]. NEC induced increase in paracellular translocation of FITC-labelled dextran was reduced by enteral HMO administration in a murine NEC model [38]. In addition, HMO administration normalizes the number of goblet cells in the intestinal villi (mucin 2 (Muc2) positive cells) that is decreased by NEC protocol exposure [38,86] and tended to increase the mRNA expression of Muc2 and trefoil factor 3 (TFF3) in NEC protocol exposed mice [38]. Interestingly, the effect of enteral HMO treatment on goblet cell numbers was abolished in the presence of an inhibitor of the ER chaperone protein PDI, suggesting a mechanism behind the protective effects of HMO administration could be induction of the unfolded protein response (UPR) [38]. In a preterm pig model of NEC, enrichment of formula feeding with a mixture of four HMO did not prevent small intestinal adhesion and tissue invasion of bacteria measured with fluorescence in situ hybridization staining and did not change small intestinal mRNA expression of mucin 1 (Muc1) and Muc2 [134].

#### 3.7.3. Protein or Amino Acid-Based Feeding Interventions

Lactoferrin, lysozyme, IAP and lactadherin are the protein/amino acid-based enteral feeding interventions that have been studied in relation to barrier function in experimental models of NEC. In a preterm pig NEC model, enteral bovine lactoferrin administration was associated with increased intestinal permeability, as demonstrated by an increased lactulose mannitol ratio following a dual sugar absorption test [35]. Enteral supplementation of lysozyme in a rat NEC model resulted in a less permeable mucus barrier, as reflected by reduced effective diffusivity of amine and carboxyl modified particles, less linear movements of *E. coli* through intestinal mucus and reduced *E. coli* movement speed through intestinal mucus [131]. In addition, lysozyme supplementation lowered the amount of sialic acid in the intestinal mucus and was associated with an altered mucus structure analysed with confocal imaging and SEM [131]. Ex-vivo measurement of ileal barrier function with FITC-labelled dextran 10 kD showed enteral IAP, both in low and a high dose, prevented an increased intestinal permeability in a rat NEC model [103]. Furthermore, protein expression of claudin-1 decreased and protein expression of claudin-3 increased with IAP administration, while occludin and ZO-1 or the mRNA expression of these proteins remained unaltered [103]. Another study using enteral IAP reported reduced plasma endotoxemia at higher, but not at a low dose [160]. Lactadherin supplementation in a rat NEC model reduced leakage of FITC-labelled dextran from the intestinal lumen into the blood [51]. Furthermore, enteral lactadherin administration reduced NEC induced disruption of cell junctions, improved anchoring of TJ complexes and reduces the space between adjacent cells, as was observed with transmission electron microscopy [51]. Enteral lactadherin prevented NEC induced increase of mRNA levels for claudin-3 and Junctional Adhesion Molecule A (JAM-A) and the protein levels of claudin-3, JAM-A and E-cadherin [51]. In addition, it administration changed localization of claudin-3 towards the cell membranes and along the crypt-villus junction, which was also seen in the dam fed control group [51]. Localization of occludin was also normalized by lactadherin treatment, as in the control group it was predominantly expressed at the cell membranes along the villus. E-cadherin localization of E-cadherin was also changed by lactadherin treatment [51]. No differences in JAM-A localization were found in NEC or lactadherin supplementation compared to controls [51].

#### 3.7.4. Hormone, Growth Factor or Vitamin-Based Feeding Interventions

Various hormone and growth factor-based enteral feeding interventions have been shown to improve intestinal barrier function in experimental models of NEC. Rat EGF reduced paracellular intestinal permeability, measured with blood levels and kidney levels of [^3^H]lactulose after oral administration [137]. Transcellular permeability was not affected by the NEC protocol or EGF treatment [137]. In addition, ileal mRNA and protein levels of occludin and claudin-3 and jejunum mRNA and protein levels of claudin-3 were reduced by EGF treatment to dam fed control levels and occludin and claudin-3 in the ileum were redistributed towards the apical and basolateral membranes along the crypt-villus axis contributing to a functional TJ barrier [137]. JAM-A and ZO-1 were more markedly/sharply expressed on immunofluorescence pictures following oral recombinant EGF from soybean administration, probably indicating better incorporation in TJ complexes of these proteins [104]. Enteral EGF treatment of NEC protocol exposed mice also significantly increased the number of goblet cells (Muc2) in the ileum and thickened the villus mucus layer compared to both NEC protocol exposed and control mice [137]. In addition, ileal mRNA level of Muc2 was increased by EGF treatment in rat and mouse models of NEC [37,137]. Importantly, an increased mRNA expression of mouse atonal homolog 1 (Math1), a transcription factor that is important for secretory cell lineage differentiation, was found in both ileum and jejunum upon EGF treatment, suggesting enteral EGF promotes goblet cell maturation and differentiation [137]. Finally, SEM of ileal goblet cells showed normalization of the goblet cell phenotype that was disturbed in NEC animals by EGF treatment, with mucin droplets on the outer cell surface [137]. In both rat [58,62,105] and mouse [61,175] NEC models, in which intestinal permeability was measured by administration of oral 73 kD FITC-labelled dextran, intestinal permeability was considerably reduced by HB-EGF treatment, both at 48 h [61,62,105], 72 h [62,105] and 96 h [175] after birth. Enteral HB-EGF administration significantly increased ileal mRNA levels of Muc2 compared to both NEC protocol exposed and dam fed animals [37]. Another study reported enteral administration of HB-EGF prevented a loss of goblet cells (alcian blue/periodic acid–Schiff (AB-PAS)) in the jejunum of NEC protocol stressed rats [176]. In addition, bacterial adherence to intestinal villi in experimental NEC was prevented by HB-EGF addition to formula feeding in a rat NEC model [59]. The effects of enteral administration of erythropoietin (EPO) on intestinal barrier function in NEC were assessed in a rat NEC model [66]. Paracellular intestinal permeability, measured with a FITC-labelled dextran 10 kD assay, was almost completely reduced to control levels by enteral EPO administration [66]. In addition, EPO administration prevented loss of ZO-1 in the TJ of histological normal ileal villi from NEC exposed animals. EPO treatment, however, did not alter claudin-1, claudin-3, E-cadherin or β-catenin protein levels in experimental NEC. It was shown the effects of EPO on the intestinal barrier function may be PI3k/Akt signaling pathway related [66]. Interestingly, in the same study, enteral administration of TGF-β failed to protect the intestinal barrier function and did not activate Akt [66]. Administration of IGF1 prevented a decrease in Muc2 protein levels at 24 h in NEC protocol exposed rats and induced an increase in Muc2 protein level at 72 h compared to control and NEC protocol exposed animals [65]. In addition, IGF1 prevented a NEC protocol decrease in secretory IgA levels at 72 h, but not at 24 h and 48 h [65].

In contrast to hormones and growth factors, evidence for vitamin driven effects on the intestinal barrier is scarce; Enteral vitamin A administration increased the intestinal protein expression of the TJ proteins claudin-1, occludin and ZO-1 in a murine NEC model [111].

#### 3.7.5. Probiotic Feeding Interventions

Many probiotic feeding interventions can improve intestinal barrier functions in the context of NEC. Administration of a *Bifidobacterium* mixture in a rat NEC model increased ileal protein and mRNA expression of β defensin 2 [115]. Daily orogastric administration of *Bifidobacterium infantis* reduced endotoxemia by 10-fold at 48 h in a rat NEC model. In contrast, no differences were seen when the intestinal barrier function was assessed with an oral FITC-labelled dextran assay at 8 h, 24 h or 48 h [75]. The authors suggested that *Bifidobacterium infantis* may protect TJ, thereby preventing bacterial transfer, whereas mucosal barrier loss leading to FITC-labelled dextran leakage could be dependent on other mechanisms such as apoptosis that were not inhibited by *Bifidobacterium infantis* [75]. In a murine NEC model, enteral administration of *Bifidobacterium infantis* prior to NEC induction partially prevented internalisation of claudin-4 into the enterocyte cytoplasm and preserved claudin-4 protein expression, occludin presence at the TJ complex and co-fractionation of claudins-2 and -4 and the membrane lipid-raft protein caveolin 1. Moreover, in this study, *Bifidobacterium infantis* administration reduced intestinal permeability as measured with an oral FITC-dextran assay [116]. *Bifidobacterium bifidum* prevented a NEC induced increase in the TJ proteins occludin and claudin-3 and normalized the cellular distribution and localization of these proteins, suggesting enhanced development and formation of functional TJ in a rat model [73]. In addition, although protein levels did not change, cellular distribution and localization of adherence junctions α-catenin, β-catenin and E-cadherin were partially normalized towards the situation in dam fed animals [73]. In the same study, enteral administration of *Bifidobacterium bifidum* further reduced the ileal Muc2 mRNA expression in NEC exposed animals and did not prevent NEC induced reduction of Muc2-positive cells [73]. On the other hand, *Bifidobacterium bifidum* treatment partially prevented NEC induced increase of mucin 3 (Muc3) mRNA expression. TFF3 was not affected by either NEC or Bifidobacterium bifidum treatment on mRNA level, but on protein level NEC protocol exposed animals showed an increase in TFF3-positive cells that was completely prevented by *Bifidobacterium bifidum* [73]. Ileal mRNA expression of ZO-1, claudin-1 and occluding were reduced (normalization towards breast fed controls) by enteral administration of *Bifidobacterium breve* M-16V in a rat NEC model [117]. Pre-treatment with *Bacteroides fragilis* strain ZY-312 before *Cronobacter sakazkii* induced NEC improves the intestinal barrier function (FITC-labelled dextran 4 kD assay) and increases the ZO-1 expression compared to NEC protocol exposed rats that were not pre-treated [113]. In addition, intestinal protein levels of IgA were increased following *Bacteroides fragilis* pre-treatment compared to NEC protocol exposed animals [113]. Enteral administration of *Lactobacillus reuteri* biofilms on unloaded [71], MRS loaded microspheres [71], sucrose loaded microspheres [72] and maltose loaded microspheres [72], but not administration of unbound *Lactobacillus reuteri* [71,72], improved intestinal barrier function measured by a functional orogastric FITC-dextran assay in a rat NEC model. Finally, *Lactobacillus rhamnosus* GG reduced NEC-protocol induced mucosal infiltration of bacteria following enteral administration [161].

#### 3.7.6. Other Enteral Feeding Interventions

Oral supplementation of bovine milk exosomes prevented NEC induced decrease of goblet cells (AB-PAS, Muc2) in mice [118]. The number of cells positive for GRP94, an ER chaperone protein that has a crucial role in goblet cell maintenance and a co-receptor for Wnt signaling was reduced in mice exposed to a NEC protocol, however, this was largely prevented by bovine milk exosome administration [118]. In addition, human breast milk exosomes partially prevented NEC induced reduction of goblet cells (Muc2) and Muc2 mRNA expression upon enteral administration in a mouse NEC model [119]. Enteral administration of berberine increased ileal protein levels of Muc2 and secretory IgA [79]. Finally, enteral human β-defensin-3 preserved ZO-1 protein expression that was lost by exposure to the NEC inducing protocol in a rat NEC model [81].

**Table 8 nutrients-13-01726-t008:** Effect of enteral feeding interventions that improve intestinal barrier function in experimental animal models of NEC.

Enteral Feeding Intervention	Effect on Intestinal Barrier Function (Compared to NEC Protocol Exposure without Feeding Intervention)
**Fat-based interventions**	
PUFA	Endotoxemia (plasma) ↓ [45]
DHA	Ileal effective diffusivity amine modified particles ↓ [131] Ileal effective diffusivity carboxyl modified particles ↓ [131] Ileal linear movements *E. coli* through intestinal mucus ↓ [131] Ileal movement speed *E. coli* through intestinal mucus ↓ [131] Ileal sialic acid content mucus ↓ [131] Ileal mucus structure (confocal imaging/SEM) = [131]
**Carbohydrate/sugar-based interventions**	
Hyaluronan 35 kD	Intestinal permeability (functional orogastric FITC-dextran assay) ↓ [94] Small intestinal occludin (protein) ↑ [94] Small intestinal claudin-4 (protein) ↑ [94] Small intestinal claudin-3 (protein) ↑ [94] Small intestinal claudin-2 (protein) ↑ [94] Small intestinal ZO-1 (protein) ↑ [94] Small intestinal occludin localization [94] Small intestinal claudin-3 localization [94]
HMO	Intestinal permeability (functional orogastric FITC-dextran assay) ↓ [38] Ileal number Muc2-positive cells ↑ [38,86] Ileal Muc2 (mRNA) ↑ (trend) [38] Ileal TFF3 (mRNA) ↑ (trend) [38]
Mixture of four HMOs	Small intestinal bacterial adhesion and tissue invasion = [134] Small intestinal Muc1 (mRNA) = [134] Small intestinal Muc2 (mRNA) = [134]
**Protein/amino acid-based interventions**	
IAP	Ileal intestinal permeability (ex-vivo FITC-dextran assay) ↓ [103] Ileal claudin-1 (protein) ↓ [103] Ileal claudin-3 (protein) ↑ [103] Ileal occludin (protein) = [103] Ileal ZO-1 (protein) = [103] Ileal claudin-1 (mRNA) = [103] Ileal claudin-3 (mRNA) = [103] Ileal occludin (mRNA) = [103] Ileal ZO-1 (mRNA) = [103] Endotoxemia (plasma) ↓ [160]
Bovine lactoferrin	Lactulose/mannitol recovery ratio in urine ↑ (only in animals with NEC) [35]
Lysozyme	Ileal effective diffusivity amine modified particles ↓ [131] Ileal effective diffusivity carboxyl modified particles ↓ [131] Ileal linear movements e coli through intestinal mucus ↓ [131] Ileal movement speed e coli through intestinal mucus ↓ [131] Ileal sialic acid content mucus ↓ [131] Ileal mucus structure (confocal imaging/SEM) changed [131]
Lactadherin	Intestinal permeability (ex-vivo FITC-dextran assay) ↓ [51] Ileal organization of cell junctions, anchoring of the TJ complexes and space between adjacent Cells improved (transmission electron microscropy) [51] Ileal claudin-3 (mRNA) ↓[51] Ileal JAM-A (mRNA) ↓ [51] Ileal claudin-3 (protein) ↓ [51] Ileal JAM-A (protein) ↓ [51] Ileal E-cadherin (protein) ↓ [51] Ileal claudin-3 distribution towards cell membranes along crypt-villus junction (normalization) [51] Ileal occludin distribution towards cell membranes along villus (normalization) [51] Ileal E-cadherin distribution towards cell membranes of villus and basolateral region of crypt cells [51] Ileal JAM-A distribution = [51]
**Hormone/growth factor/vitamin- based interventions**	
EGF	Paracellular intestinal permeability (functional orogastric [3*H*]lactulose assay) ↓ [137] Transcellular intestinal permeability (functional orogastric [3*H*]rhamnose assay) = [137] Ileal occludin (mRNA) ↓ [137] Jejunal and ileal claudin-3 (mRNA) ↓ [137] Ileal occludin (protein) ↓ [137] Jejunal and ileal claudin-3 (protein) ↓ [137] Ileal occludin distribution towards apical and basolateral membrane of crypt-villus axis [137] Ileal claudin-3 distribution towards apical and basolateral membrane of crypt-villus axis [137] Ileal number of goblet cells (Muc2 protein) ↑ [137] Ileal mucus layer on top villi tips ↑ [137] Ileal Muc2 (mRNA) ↑ [137] Jejunal and ileal Math1 (mRNA) ↑ [137] Ileal goblet cell phenotype normalized (scanning electron microscopy) [137]
Recombinant EGF from soybean extract	Ileal ZO-1 more sharply expressed, better incorporation in TJ (IF) [104] Ileal JAM-A more sharply expressed, better incorporation in TJ (IF) [104]
HB-EGF	Intestinal permeability (functional orogastric FITC-dextran assay) ↓ [58,61,105,175] Ileal Muc2 (mRNA) ↑ [37] Jejunal goblet cell number (AB/PAS) ↑ [176] Ileal bacterial adherence to intestinal villi ↓ (scanning electron microscope) [59]
IGF1	Ileal secretory IgA (protein) ↑ [65] Ileal Muc2 (protein) ↑ [65]
EPO	Intestinal permeability (functional orogastric FITC-dextran assay) ↓ [66] Intestinal ZO-1 loss from TJ intact villi (protein) ↓ [66] Intestinal caudin-1 (protein) = [66] Intestinal caudin-3 (protein) = [66] Intestinal E-cadherin (protein) = [66] Intestinal β-catenin (protein) = [66] Intestinal p-Akt (protein) ↑ [66]
Vitamin A	Intestinal claudin-1 (protein) ↑ [111] Intestinal occludin (protein) ↑ [111] Intestinal ZO-1 (protein) ↑ [111]
**Probiotic interventions**	
*Bifidobacterium* mixture	Ileal β defensin (protein) ↑ [115] Ileal β defensin (mRNA) ↑ [115]
*Bifidobacterium infantis*	Endotoxemia (plasma) ↓ [75] Intestinal permeability (functional orogastric FITC-dextran assay) ↓ [116] Small intestinal internalization of claudin-4 in enterocyte cytoplasm (protein) ↓ [116] Small intestinal claudin-4 expression in TJ complex (protein) ↑ [116] Small intestinal occludin expression in TJ complex (protein) ↑ [116] Small intestinal co-fractioning of claudins-2 and -4 and caveolin 1 (protein) ↑ [116] intestinal permeability (functional orogastric FITC-dextran assay) ↓ [116]
*Bifidobacterium bifidum* OLB6378	Ileal occludin (protein) ↓ [73] Ileal claudin-3 (protein) ↓ [73] Ileal occludin distribution towards crypts (normalization) [73] Ileal claudin-3 distribution towards crypts and cell membrane (normalization) [73] Ileal α-catenin (protein) = [73] Ileal β-catenin (protein) = [73] Ileal e-cadherin (protein) = [73] Ileal α-catenin distribution towards complete villus length and cell membrane (normalization) [73] Ileal β-catenin distribution towards complete villus length except for villi tips and cell membrane (normalization) [73] Ileal e-cadherin distribution towards crypts and cell membrane (normalization) [73] Ileal muc2 (mRNA) ↓ [73] Ileal Muc3 (mRNA) ↓ [73] Ileal TFF3 (mRNA) = [73] Ileal number of goblet cells (Muc2 protein) = [73] Ileal number of TFF3 positive cells (TFF3 protein) ↓ [73]
*Bifidobacterium breve* M-16V	Ileal ZO-1 (mRNA) ↓ [117] Ileal claudin-1 (mRNa) ↓ [117] Ileal occludin (mRNA) ↓ [117]
*Bacteroides fragilis* strain ZY-312	Intestinal permeability (functional orogastric FITC-dextran assay) ↓ [113] Intestinal ZO-1 (protein) ↑ [113] Intestinal secretory IgA (protein) ↑ [113]
*Lactobacillus reuteri* DSM 20016	Intestinal permeability (functional orogastric FITC-dextran assay) = [71,72]
*Lactobacillus reuteri* biofilm on unloaded microspheres	Intestinal permeability (functional orogastric FITC-dextran assay) ↓ [71]
*Lactobacillus reuteri* biofilm on MRS loaded microspheres	Intestinal permeability (functional orogastric FITC-dextran assay) ↓ [71]
*Lactobacillus reuteri* biofilm on sucrose loaded microspheres	Intestinal permeability (functional orogastric FITC-dextran assay) ↓ [72]
*Lactobacillus reuteri* biofilm on maltose loaded microspheres	Intestinal permeability (functional orogastric FITC-dextran assay) ↓ [72]
*Lactobacillus rhamnosus* GG	Colonic mucosal infiltration of bacteria (EUB338 staining) ↓ [161]
**Other interventions**	
Bovine milk exosomes	Distal ileal number of goblet cells (Muc2 protein) ↑ [118] Distal ileal number of goblet cells (AB-PAS) ↑ [118] Distal ileal number of GRP93 positive cells (protein) ↑ [118]
Human breast milk exosomes	Distal ileal number of goblet cells (Muc2 protein) ↑ [119] Distal ileal Muc2 (mRNA) ↑ [119]
Berberine	Distal ileal Muc2 (protein) ↑ [79] Distal ileal secretory IgA (protein) ↑ [79]
Human β-defensin-3	Terminal ileal ZO-1 (protein) ↑ [81]

↑ depicts an increase, ↓ depicts a decrease; PUFA, polyunsaturated fatty acids; DHA, docosahexaenoic acid; HMO, human milk oligosaccharides; IAP, intestinal alkaline phosphatase; EGF, epidermal growth factor; HB-EGF, hemoglobin-binding EGF-like growth factor; EPO, erythropoietin.

### 3.8. NEC Pathophysiology: Vascular Dysfunction, Hypoxia-Ischemia and Free Radical Formation

Intestinal microvasculature alterations, hypoxia, ischemia and oxidative stress (increased reactive oxygen and nitrogen species (together called ROS)) are important factors contributing to NEC pathogenesis. In physiological conditions, intestinal vasodilatation counterbalances effects of vasoconstriction, thereby facilitating appropriate intestinal blood supply [177]. During NEC, the balance between vasodilatation and vasoconstriction is disturbed, leading to hypoxia, ischemia and ROS formation. In premature neonates, increased vascular resistance in the superior mesenteric artery (measured with Doppler flow velocimetry) was associated with an increased risk of developing NEC [178]. An important intestinal vasodilator that has been studied intensively in the context of NEC is NO. NO is synthesized from arginine by NOS. NOS has three isoforms of which inducible NOS (iNOS) and endothelial NOS (eNOS) are of importance for NEC pathogenesis. eNOS is naturally expressed in the intestinal vasculature and provides background levels of NO [177]. In tissue from infants with NEC it was found that although eNOS protein expression was not reduced during NEC, eNOS function was hampered [179]. In contrast to the protective effects of low levels of NO derived from eNOS, excessive NO production by iNOS seems to contribute to NEC pathogenesis [177,180]. iNOS has been observed to be upregulated in the enterocytes of infants with NEC [181]. NO or reactive species derived from NO have been implied to suppress intestinal oxygen consumption [182] and inhibit enterocyte proliferation and migration [177,180]. Moreover, they increase gut barrier permeability by affecting TJ and gap junctions or inducing enterocyte apoptosis and necrosis [177,180,183]. In addition to changes in vasodilators, higher concentrations of the potent vasoconstrictor endothelin-1 (ET-1) and vasoconstriction are found in diseased parts of the intestine resected from NEC patients when compared with relatively healthy parts of the same resected gut [184]. Of importance, several inflammatory mediators, have been shown to influence vascular tone via vasoconstrictors and vasodilators; for instance, PAF increases ET-1 mediated vasoconstriction and thereby contributes to impaired blood flow in NEC [177].

### 3.9. Enteral Feeding and Vascular Dysfunction, Hypoxia-Ischemia and Free Radical Formation in Animal Models of NEC

Several studies have described the effect of enteral feeding interventions on either ROS, iNOS expression, antioxidant capacity or intestinal vasculature in animal models of NEC (Table 9).

#### 3.9.1. Fat-Based Feeding Interventions

Fat-based dietary interventions may reduce oxidative stress in the context of NEC. iNOS mRNA expression was not altered by enteral administration of PUFA with or without nucleotides [45]. However, pre-digested or very low-fat formula feeding reduced intestinal lipid accumulation and accumulation of ROS in the distal ileum of NEC-protocol exposed mice [85]. In addition, both diets reduced intestinal malonaldehyde (MDA) protein levels, indicating reduced lipid oxidation [85]. Enteral administration of MFGM in a rat NEC model lowered ileum iNOS mRNA expression and MDA protein levels and prevented a NEC induced decrease of antioxidant enzyme superoxide dismutase (SOD) protein levels [48].

#### 3.9.2. Carbohydrate or Sugar-Based Feeding Interventions

HMO have been shown to positively influence blood flow and reduce oxidative stress in experimental NEC. Enteral administration of the HMO 2′-FL increased mesenteric blood flow as measured with mesenteric micro-angiography to the levels of breast-fed mice in a murine NEC model [91]. This effect was mediated through preserved eNOS expression and function [91] and reduced intestinal iNOS mRNA expression [91]. In both a murine and pig model of NEC, 2′-FL, 6′-SL and a combination of 2′-FL and 6′-SL reduced intestinal 3′-nitrotyrosine levels, a marker for nitrogen free radical species, indicating reduced oxidative stress [92]. GOS/FOS administration increased terminal ileum mRNA expression of the anti-oxidant enzymes SOD-1 [135], SOD-3 [136], glutathione peroxidase (GSH-Px)-1 [135], GSH-Px-7 [135] and catalase (CAT) [135] in a rat NEC model.

#### 3.9.3. Protein or Amino Acid-Based Feeding Interventions

Mainly amino acid-based feeding interventions have been shown to influence oxidative stress. Enteral supplementation of both l-carnitine and l-arginine normalized the level of thiobarbituric acid reactive substances, suggesting reduced lipid peroxidation and/or increased antioxidant activity in a murine NEC model [99]. However, antioxidant enzymes tissue SOD and CAT activity was not altered by either l-carnitine or l-arginine supplementation [99]. Although l-arginine supplementation also increased nitrate levels (stable metabolite of NO), this was not statistically significant compared to untreated NEC protocol exposed animals [99]. Intestinal hypoxia, as evaluated by pimonidazole staining, was reduced by enteral supplementation of arginine in a murine NEC model [98]. This effect was probably mediated by improved blood flow following increased vasodilatation, as arginine supplementation to formula increased postprandial arterial diameter in the intestinal microcirculation [98]. Addition of N-acetylcysteine to standard formula reduced both ROS levels and lipid peroxidation (MDA) in the terminal ileum of NEC-protocol exposed mice [85]. Glutamine administration did not reduce terminal ileum nitric oxide production in a rat NEC model [96]. Enteral IAP administration inhibits ileal iNOS mRNA expression, both in high and lower dosages, in a rat NEC model [160]. In addition, enteral IAP dose dependently decreased ileal levels of nitrogen free radical species [160].

#### 3.9.4. Hormone or Growth Factor-Based Feeding Interventions

Soybean-derived recombinant human EGF reduced ileal iNOS mRNA levels upon enteral supplementation in a rat NEC model [104]. An elegant study by Yu et al. in a rat NEC model observed that enteral HB-EGF administration preserved villus microvascular blood flow, prevented NEC induced changes in intestinal villus microvascular structure and significantly increased submucosal intestinal blood flow [106]. In addition, oral administration of the hormone relaxin increases ileal blood flow measured by laser Doppler flowmetry in a rat NEC model [107]. In a mouse NEC model, enteral vitamin D administration decreased MDA protein expression (reduced lipid oxidation) and increased GSH-Px protein expression [112].

#### 3.9.5. Probiotic Feeding Interventions

Several probiotic interventions effectively reduce oxidative stress in experimental NEC. *Lactobacillus rhamnosus* supplementation (both alive and dead) as well as supplementation of *Lactobacillus rhamnosus* isolated microbial DNA reduced terminal ileum mRNA expression of iNOS in a murine NEC model and in a premature piglet NEC model [39]. This effect is likely mediated through TLR9 signaling, as it was not observed in TLR9 knock-down animals [39]. Also oral administration of CpG-DNA, a ligand of TLR9 signaling, reduced terminal ileum iNOS mRNA levels in mice [39]. *Bacteroides fragilis* strain ZY-312 prevents *Cronobacter sakazakii* induced iNOS induction in a rat NEC model [113]. *Lactobacillus reuteri* DSM17938 administration increased SOD activity, SOD inhibition rate and glutathione (GSH) protein levels while decreasing glutathione disulphide (GSSG) protein levels, MDA protein levels and the GSSG/GSH ratio, suggesting improved antioxidant capacity and reduced oxidative stress [70]. In a rat NEC model, enteral administration of *Saccharomyces boulardii* increased the mRNA expression of SOD-1 [135], SOD-3 [136], GSH-Px-1 [135], GSH-Px-3 [135], GSH-Px-4 [135], GHS-Px-7 [135] and CAT [135] in the terminal ileum.

#### 3.9.6. Other Enteral Feeding Interventions

Oral treatment with ginger increased intestinal protein levels of the antioxidant enzymes SOD and GSH-Px and reduced protein levels of the oxidative stress markers MDA and xanthine oxidase (XO) [121]. Intestinal MDA protein levels were also significantly reduced by oral sesamol treatment, concomitant with increased SOD protein levels [125]. In addition, levels of the GSH-Px were increased, without reaching statistically significance [125]. Enteral treatment with fennel seed extracts in a rat model of NEC decreased the intestinal total oxidant status, the oxidative stress index, the amount of advanced oxidation protein products and the concentration of lipid hydroperoxide and 8-hydroxydeoxyguanosine (oxidized guanine, 8-OhdG), while increasing the total antioxidant status, indicating reduced oxidative stress [122]. Addition of rat amniotic fluid to formula feeding reduced intestinal mRNA levels of iNOS in a rat NEC model [63]. Enteral administration of amniotic fluid also reduced distal small intestinal iNOS mRNA levels in a preterm pig model of NEC [123] and terminal ileum iNOS protein and mRNA expression in a mouse model of NEC [40]. Interestingly, these effects may be EGFR signaling mediated, as the effects on iNOS expression were largely lost with co-administration of the EGFR inhibitor cetuximab or with amniotic fluid depleted of EGF [40]. Berberine administration reduced ileal iNOS mRNA expression in a rat NEC model [79]. In addition, in a rat NEC model, oral administration of the flavonoid astragaloside IV dose dependently increased distal ileum protein concentrations of GSH and SOD, while decreasing protein levels of MDA, indicating reduction of oxidative stress by astragaloside IV [126]. Finally, enteral supplementation with resveratrol, a polyphenol produced by plants, prevented a NEC induced increase in ileal iNOS protein expression in a rat NEC model [127].

**Table 9 nutrients-13-01726-t009:** Effect of enteral feeding interventions that reduce vascular dysfunction, hypoxia and free radical formation in experimental animal models of NEC.

Enteral Feeding Intervention	Effect on Vascular Dysfunction, Hypoxia and Free Radical Formation (Compared to NEC Protocol Exposure without Feeding Intervention)
**Fat-based interventions**	
PUFA	Intestinal iNOS (mRNA) = [45]
Pre-digested fat (less long chain triacylglycerol, not dependent on intestinal lipases	Ileal ROS accumulation ↓ (DHE staining) [85] Ileal MDA (protein) ↓ [85]
Very low-fat diet	Ileal ROS accumulation ↓ (DHE staining) [85] Ileal MDA (protein) ↓ [85]
MFGM	Ileal iNOS (mRNA) ↓ [48] Intestinal MDA (protein) ↓ [48] Intestinal SOD (protein) ↑ [48]
**Carbohydrate/sugar-based interventions**	
2′-FL	Mesenteric blood flow ↑ (mesenteric micro-angiography) (eNOS dependent) [91] Intestinal iNOS (mRNA) ↓ [91] Small intestinal free nitrogen species, 3-nitrotyrosine (protein) ↓ [92]
6′-SL	Small intestinal free nitrogen species, 3-nitrotyrosine (protein) ↓ [92]
2′-FL + 6′-SL	Small intestinal free nitrogen species, 3-nitrotyrosine (protein) ↓ [92]
GOS/FOS	Terminal ileum SOD-1 (mRNA) ↑ [135] Terminal ileum SOD-3 (mRNA) ↑ [136] Terminal ileum GSH-Px-1 ↑ [135] Terminal ileum GSH-Px-7 ↑ [135] Terminal ileum CAT ↑ [135]
**Protein/amino acid-based interventions**	
L-Arginine	Intestinal thiobarbituric acid reactive substances ↓ [99] Intestinal SOD (protein) = [99] Intestinal CAT (protein) = [99] Intestinal nitrate (stable metabolite of NO) ↑ (NS) [99] Intestinal hypoxia ↓ (pimonidazole) [98] Postprandial arterial diameter intestinal microcirculation ↑ [98]
L-Carnitine	Intestinal thiobarbituric acid reactive substances ↓ [99] Intestinal SOD (protein) = [99] Intestinal CAT (protein) = [99]
Glutamine	Terminal ileal NO production = [96]
IAP	Ileal iNOS (MMA) ↓ [160] Ileal free nitrogen species, 3-nitrotyrosine (protein) ↓ [160]
N-Acetylcysteine	Ileal ROS accumulation ↓ (DHE staining) [85] Ileal MDA (protein) ↓ [85]
**Hormone/growth factor/Vitamin-based interventions**	
Recombinant EGF from soybean extract	ileal iNOS (mRNA) ↓ [104]
HB-EGF	Villus microvascular blood flow ↑ (angiography) [106] Villus microvascular structure preserved (angiography, scanning electron microscopy) [106] Submucosal intestinal blood flow ↑ (angiography) [106]
Relaxin	Ileal blood flow ↑ (laser Doppler flowmetry) [107]
Vitamin D	Intestinal MDA (protein) ↓ [112] Intestinal GSH-Px (protein) ↑ [112]
**Probiotic interventions**	
*Bacteroides fragilis* strain ZY-312	Intestinal iNOS (protein) ↓ [113]
*Lactobacillus rhamnosus* HN001 (alive)	Terminal ileal iNOS (mRNA) ↓ (TLR9 dependent) [39]
*Lactobacillus rhamnosus* HN001 (dead, UV-radiated)	Terminal ileal iNOS (mRNA) ↓ [39]
*Lactobacillus rhamnosus* HN001 isolated microbial DNA	Terminal ileal iNOS (mRNA) ↓ [39]
CpG-DNA	Terminal ileal iNOS (mRNA) ↓ [39]
*Lactobacillus reuteri* DSM 17938	Terminal ileal SOD activity (U/mg protein) ↑ [70] Terminal ileal SOD inhibition rate (%) ↑ [70] Terminal ileal GSSG concentration (protein) ↓ [70] Terminal ileal GSH concentration (protein) ↑ [70] Terminal ileal GSSG/GSH ratio (protein) ↓ [70] Terminal ileal MDA concentration (protein) ↓ [70]
*Saccharomyces Boulardii*	Terminal ileal SOD-1 (mRNA) ↑ [135] Terminal ileal SOD-3 (mRNA) ↑ [136] Terminal ileal GSH-Px-1 ↑ [135] Terminal ileal GSH-Px-3 ↑ [135] Terminal ileal GSH-Px-4 ↑ [135] Terminal ileal GSH-Px-7 ↑ [135] Terminal ileal CAT ↑ [135]
**Other interventions**	
Ginger	Intestinal SOD (protein) ↑ [121] Intestinal GSH-Px (protein) ↑ [121] Intestinal MDA (protein) ↓ [121] Intestinal XO (protein) ↓ [121]
Sesamol	Intestinal SOD (protein) ↑ [125] Intestinal GSH-Px (protein) ↑ (NS) [125] Intestinal MDA (protein) ↓ [125]
Fennel seed extracts	Intestinal total oxidant status (µmol H_2_O_2_ equivalent/g protein) ↓ [122] Intestinal oxidative stress index (total oxidant status/total antioxidant status) ↓ [122] intestinal advanced oxidation protein products (ng/mg protein) ↓ [122] intestinal lipid hydroperoxide (nmol/L) ↓ [122] intestinal 8-hydroxydeoxyguanosine (8-OhdG, ng/mL) ↓ [122] intestinal total antioxidant status (mmol Trolox equivalent/g protein) ↑ [122]
Amniotic fluid	intestinal iNOS (mRNA) ↓ [63] distal small intestinal/terminal ileum iNOS (mRNA) ↓ [40,123] terminal ileum iNOS (protein) ↓ [40]
Berberine	ileal iNOS (mRNA) ↓ [79]
Astragaloside IV	distal ileum GSH (protein) ↑ [126] distal ileum SOD (protein) ↑ [126] distal ileum MDA (protein) ↓ [126]
Resveratrol	ileum iNOS (protein) ↓ [127]

↑ depicts an increase, ↓ depicts a decrease; PUFA, polyunsaturated fatty acids; MFGM, milk fat globule membrane; 2′-FL, 2′-fucosyllactose; 6′-SL, 6′-sialyllactose; GOS: galacto-oligosaccharides; FOS, fructo-oligosaccharides; IAP, intestinal alkaline phosphatase; EGF, epidermal growth factor; HB-EGF, hemoglobin-binding EGF-like growth factor.

### 3.10. NEC Pathophysiology: Intestinal Epithelial Cell Death and Proliferation

Several forms of cell death can be distinguished in the intestinal epithelium including apoptosis, necrosis and necroptosis [185] and all of these mechanisms have been described in NEC pathophysiology [181,186]. Whereas necrosis is uncontrolled and comes with collateral damage, both apoptosis and necroptosis are tightly regulated by several cellular pathways [185]. Increased apoptosis is detected in the intestinal epithelium of NEC patients; increased terminal deoxynucleotidyl transferase dUTP nick end labelling (TUNEL) staining was observed in villus enterocytes in NEC biopsies [181] and mRNA an protein expression of caspase 3 and Bax were found to be increased in ileum of patients with NEC compared to controls [187]. In addition, the mRNA expression of the anti-apoptotic Bcl2 was decreased [187]. NEC is also associated with intestinal upregulated mRNA expression of the three major necroptosis pathway genes and mRNA expression of these genes positively correlates with disease severity [186]. Also on protein level increased necroptosis is detected in specimen from infants with NEC [186]. In experimental NEC (murine model), both pharmacological and genetic inhibition of necroptosis decreased intestinal epithelial cell death and mucosal inflammation, suggesting a role for necroptosis in NEC pathogenesis [186]. Last, autophagy, is observed at higher levels in NEC tissue compared to control tissue [55,188]. Autophagy is the transfer of cytoplasmic components, organelles or infectious agents to lysosomes for degradation [189]. Although this is in principle a cell survival mechanism, it ultimately lead to cell death [189]. Another mechanism that may contribute to cell death in NEC is intestinal endoplasmic reticulum (ER) stress. In tissue of a subset of patients with acute NEC splicing of the ER stress related protein X-box binding protein 1 (XBP1) was detected with concomitant increased mRNA and protein expression of ER stress markers binding immunoglobulin protein (BiP) and C/EBP homologous protein (CHOP), suggesting increased ER stress [190]. Importantly, ER stress correlated with increased morphological damage and intestinal inflammation and worse surgical outcome [190]. Finally, increased mRNA expression of spliced XBP1 is reported in combination with increased BiP protein expression and increased apoptosis in the crypts in NEC patients compared to controls [191].

Besides cell death, intestinal epithelial proliferation is changed during NEC. In gut samples from infants with NEC, reduced proliferation was observed in intestinal crypts [186]. In contrast, a study by Schaart et al. found increased proliferation in both severely and mildly damaged small intestine and colon of infants with NEC [167], indicating that NEC severity might be an important determinant herein. Vieten et al. reported loss of villus length in the small bowel of NEC patients, concomitant with an increased crypt depth suggesting hyperplasia and increased numbers of proliferating cells in the remaining viable crypts in both small intestine and colon. This suggests a compensatory proliferative response is triggered in NEC, that is insufficient to compensate the rapid mucosal damage in NEC [192]. Finally, loss of leucine-rich repeat-containing G-protein coupled receptor 5 (LGR5) positive stem cells was observed in human intestine resected from NEC patients compared to intestine resected from an aged-matched control infant with ileal atresia [176].

### 3.11. Enteral Feeding and Intestinal Epithelial Cell Death and Proliferation in Animal Models of NEC

An overview of enteral feeding interventions with cell death or proliferation as read-out is presented in Table 10.

#### 3.11.1. Fat-Based Feeding Interventions

Both PUFA and MPL were studied in relation to cell death in experimental NEC. Pre-treatment of rats with fish oil (rich in the PUFA DHA and EPA) reduced intestinal protein levels of BiP and the pro-apoptotic protein caspase 12, indicating reduced intestinal ER stress and potential protection against apoptosis [83]. However, in another study, enteral supplementation of PUFA did not reduce the level of intestinal epithelial apoptosis in experimental NEC [45]. In contrast, enteral administration of MPL, which are abundantly present in MFGM, did dose dependently decrease intestinal epithelial cell apoptosis indicated by decreased expression of the pro-apoptotic protein Bax, increased expression of the anti-apoptotic protein Bcl-2 and inhibited caspase activity (expression of caspase 9 and caspase 3 and TUNEL) [84]. Formula feeding supplemented with pomegranate seed oil normalized mean ileal villus length of NEC protocol exposed rats and increased ileal epithelial cell proliferation [47].

#### 3.11.2. Carbohydrate or Sugar Based Feeding Interventions

HMO have been shown to promote intestinal proliferation and reduce apoptosis in the context of NEC. In a mouse NEC model, orogastric administration of HMO restored the amount of cells positive for the proliferation marker Ki67 in the ileum [49,86,88], whereas this effect was not seen with supplementation with infant formula oligosaccharides [49]. In addition, loss of Sox9-positive stem cells was prevented by HMO treatment, but not by infant formula oligosaccharides [49]. In a preterm pig model of NEC, treatment with a mixture of four HMO did not change small intestinal mRNA expression of proliferating cell nuclear antigen (PCNA) [134]. Enteral administration of HMO reduced apoptosis (TUNEL) [86] and decreased ileal cleaved caspase-3 and hypoxia-inducible factor 1α (HIF1α) protein levels [88] in a murine NEC model. Both in a pig and murine NEC model, enteral administration of 2′-FL, 6′-SL and a combination of the two reduced intestinal epithelial apoptosis [92].

#### 3.11.3. Protein or Amino Acid-Based Feeding Interventions

Enteral administration of glutamine in a mouse model of NEC decreases intestinal epithelial cell apoptosis (TUNEL assay) and decreases expression of pro-apoptotic proteins Bax, caspase 9 and caspase 3 while increasing Bcl-2 protein expression (anti-apoptotic) [84]. In addition, enteral glutamine lowered caspase 3 protein expression in jejunum, ileum and colon in a rat model of NEC [97]. The potential harmful effects of nutritional interventions are demonstrated by a study of high-dose (10 g/L) lactoferrin supplementation in a preterm pig model of NEC. In this study, lactoferrin supplementation decreased villus length/crypt depth ratio, suggesting decreased proliferation or increased cell death in the intestinal epithelium [36]. In addition, the Bax/Bcl-2 ratio and HIF1α protein levels were elevated by supplementation of formula with lactoferrin, whereas protein levels of pro-caspase 3 and cleaved caspase 3 were not affected [36]. These detrimental effects are likely caused by the high dose of the lactoferrin used, as in in vitro experiments with cultured intestinal epithelial cells a high dose, but not lower doses, of bovine lactoferrin upregulated the expression of pro-apoptotic proteins and HIF1α signaling pathway proteins and downregulated that of anti-apoptotic proteins and proteins related to cell proliferation [36]. In another study using a mouse model of NEC, enteral recombinant lactoferrin administration (6 g/L) prevented a NEC protocol induced decrease in Ki67 immunoreactivity, preserved beta-catenin immunoreactivity and restored LGR5 mRNA levels in the distal ileum [101]. Together, these studies demonstrate that the dose of the nutritional intervention studied is important and should be taken into account when designing a clinical trial.

#### 3.11.4. Hormone, Growth Factor or Vitamin-Based Feeding Interventions

Effects of hormones and growth factors on intestinal epithelial proliferation and cell death have been studied extensively. Enteral EGF increased intestinal villus length through hyperplasia, but had no effect on intestinal epithelial proliferation as measured by PCNA immunoreactivity in experimental NEC [193]. In addition, EGF decreased levels of Bax [193], increased levels of Bcl-2 [193] and decreased the Bax-to-Bcl-2 ratio both on mRNA [193] and protein level [37,193]. In line, EGF markedly decreased cleaved caspase 3 immunoreactivity at the villus tips [193]. Ileal protein levels of Beclin 1 and LC3II, both important autophagy regulators, as well as the ratio between LC3II and LC3I were decreased by EGF treatment in NEC protocol exposed rats, indicating reduced autophagy [55]. This finding was supported by an increase of the autophagy substrate p62 by orogastric EGF administration [55]. Moreover, whereas typical signs of autophagy such as autophagosomes, autophagolysosomes and vacuoles were present in only NEC protocol exposed animals, these structural abnormalities were virtually absent in NEC protocol exposed animals that were treated with enteral EGF [55]. Enteral HB-EGF decreased intestinal TUNEL score and cleaved caspase 3 score in a rat NEC model, indicating enteral HB-EGF treatment reduces intestinal epithelial apoptosis [60]. However, in another study the Bax-to-Bcl-2 protein ratio was unaltered [37]. Enteral HB-EGF improved bromodeoxyuridine (BrdU)-positive cell migration along the crypt-villus axis [59,61] and increased intestinal epithelial proliferation (number of BrdU-positive cells) in experimental NEC [59]. In addition, in a mouse NEC model, enteral HB-EGF increased the small intestinal mRNA levels of integrin subunits α5 and β1 (but not integrin subunits α1, α2, α3 or α6) and the protein concentrations of integrin subunits α5 and β1 that were reduced by the NEC inducing protocol [61]. Orogastric HB-EGF administration increased proliferation of crypt epithelial cells that was reduced by NEC protocol exposure and prevents reduction of the number of enterocytes per villus in the jejunum of rats subjected to an experimental NEC model [176]. In addition, the number of LGR5+/prominin-1+ stem cells was significantly increased by HB-EGF administration in NEC protocol exposed rats [176].

In the small intestine of NEC protocol exposed animals without intestinal necrosis, Beclin 1 and LC3 immunoreactivity and Beclin 1 and LC3II protein levels were decreased and p62 immunoreactivity and protein levels were increased in EPO treated animals compared to non-EPO treated animals [194]. In addition, cleaved caspase 3 immunoreactivity was reduced and Bcl-2 protein levels were increased by orogastric EPO exposure [194]. In vitro evidence from an IEC-6 cell line suggests the found effects on autophagy and apoptosis are mediated through Akt/mTOR and MAPK/ERK signaling pathways respectively [194].

Evidence on the effect of vitamins on intestinal cell death and proliferation are sparse. One study investigating the effects of enteral ATRA administration found decreased levels of apoptosis in the terminal ileum intestinal crypts and preserved proliferative capacity of crypt intestinal epithelial cells in NEC protocol exposed mice [109]. In addition, vitamin D was shown to reduce cleaved caspase 3 protein expression, whereas Bcl-2 and Ki67 protein expression were increased, suggesting reduced apoptosis and increased proliferation [112].

#### 3.11.5. Probiotic Feeding Interventions

The only probiotic feeding interventions studied in relation to intestinal cell death are *Bacteroides fragilis*, *Lactobacillus rhamnosus*, *Bifidobacterium bifidum* and *Bifidobacterium breve*. Pre-treatment with *Bacteroides fragilis* strain ZY-312 lowered intestinal protein levels of caspase 3 and Bax and increased protein levels of Bcl-2 in a *Cronobacter sakazakii*-induced rat NEC model, indicating *Bacteroides fragilis* modulates apoptosis upon enteral administration [113]. In addition, treatment with *Bacteroides fragilis* reduced NEC-protocol induced inflammasome expression and pyroptosis, as demonstrated by reduced protein levels of NLRP3 inflammasome proteins (caspase-1, ASC and NLRP3), IL1β and gasdermin-D [113]. *Lactobacillus rhamnosus* GG administration partially prevents intestinal apoptosis in a mouse NEC model [161]. *Bifidobacterium bifidum* administration in a rat NEC model decreased ileal protein levels of Bax, increased protein levels of Bcl-w, reduced the Bax/Bcl-2 ratio and decreased the number of apoptotic cells (CC3-positive cells) [74]. This effect seems to be COX-2 mediated as ileal COX-2 immunoreactivity and prostaglandin E2 concentrations were upregulated by *Bifidobacterium bifidum* treatment and simultaneous administration of a COX-2 inhibitor abolished the observed reduction of apoptosis [74]. Last, supplementation of formula feeding with *Bifidobacterium breve* M-16V in a rat NEC model reduced the ileal mRNA expression of caspase 3 [117].

#### 3.11.6. Other Enteral Feeding Interventions

A broad range of other enteral feeding interventions has been shown to reduce intestinal cell death and promote proliferation in experimental models of NEC. Administration of amniotic fluid in a mouse NEC model restored terminal ileum epithelial proliferation (PCNA immunoreactivity) in a largely EGFR dependent manner [40]. Enteral ginger treatment in NEC protocol exposed rats decreased TUNEL-positive, caspase 3-positive and caspase 8-positive cell numbers and decreased caspase 3 protein levels, indicated reduced apoptosis [121]. Administration of fennel seed extracts decreased the amount of caspase 3-, caspase 8- and caspase 9-positive cells in the terminal ileum and decreased intestinal caspase 3 protein levels [122]. Supplementation of formula feeding with preterm human milk exosomes prevented NEC-protocol induced reduction in enterocyte proliferation in a rat NEC model [120]. The number of Bcl-2- and caspase 3-positive cells were significantly decreased in the intestine of NEC protocol exposed rats that were orally treated with sesamol compared to non-treated rats [125]. Enteral administration of curcumin in a rat NEC model decreased intestinal protein and mRNA expression of caspase 1 and NLRP3 in a SIRT1 mediated fashion, suggesting curcumin reduces pyroptosis [124].

**Table 10 nutrients-13-01726-t010:** Effect of enteral feeding interventions that decrease intestinal epithelial cell death and increase proliferation in experimental animal models of NEC.

Enteral Feeding Intervention	Effect on Intestinal Epithelial Cell Death and Proliferation (Compared to NEC Protocol Exposure without Feeding Intervention)
**Fat-based interventions**	
Fish oil (rich in n-3 PUFA)	Small intestinal BiP (protein) ↓ [83] Small intestinal caspase 12 (protein) ↓ [83]
PUFA	Intestinal apoptosis (TUNEL) = [45]
MPL	Small intestinal apoptosis (TUNEL) ↓ [84] Small intestinal Bax (protein) ↓ [84] Small intestinal Bcl-2 (protein) ↑ [84] Small intestinal caspase 9 (protein) ↓ [84] Small intestinal caspase 3 protein) ↓ [84]
Pomegranate seed oil	Mean ileal villus length ↑ [47] Ileal epithelial cell proliferation (PCNA) ↑ [47]
**Carbohydrate/sugar-based interventions**	
HMO	(Terminal) ileal Ki67-positive cells ↑ [49,86,88] Ileal Sox9-positive cells ↑ [49] Terminal ileal TUNEL (protein) ↓ [86] Ileal cleaved caspase 3 (protein) ↓ [88] Ileal HIF1α (protein) ↓ [88]
Mixture of four HMOs	Small intestinal PCNA (mRNA) = [134]
2′-FL	Small intestinal apoptosis (TUNEL) ↓ [92]
6′-SL	Small intestinal apoptosis (TUNEL) ↓ [92]
2′-FL + 6′-SL	Small intestinal apoptosis (TUNEL) ↓ [92]
**Protein/amino acid-based interventions**	
Lactoferrin	Proximal intestinal villus length/crypt depth ratio ↓ [36] Middle intestinal Bax-to-Bcl-2 ratio (protein) ↑ [36] Middle intestinal HIF-1α (protein) ↑ [36] Middle intestinal pro-caspase 3 (protein) = [36] Middle intestinal CC3 (protein) = [36] Distal ileal Ki67 (protein) ↑ [101] Distal ileal β-catenin (protein) ↑ [101] Distal ileal LGR5 (mRNA) ↑ [101]
L-Glutamine/glutamine	Small intestinal apoptosis (TUNEL) ↓ [84] Small intestinal Bax (protein) ↓ [84] Small intestinal Bcl-2 (protein) ↑ [84] Small intestinal caspase 9 (protein) ↓ [84] Small intestinal caspase 3 (protein) ↓ [84] Jejunum, ileum and colon caspase 3 (protein) ↓ [97]
**Hormone/growth factor/vitamin-based interventions**	
EGF	Ileal villus length ↑ [193] Ileal epithelial proliferation (PCNA) = [193] Ileal Bax (mRNA) ↓ [193] Ileal Bax (protein) ↓ [193] Ileal Bcl-2 (mRNA) ↑ [193] Ileal Bcl-2 (protein) ↑ [193] Ileal Bax-to-Bcl-2 ratio (mRNA) ↓ [193] Ileal CC3 villus tips (protein) ↓ [193] Ileal Bax-to-Bcl-2 ratio (protein) ↓ [37,193] Ileal Beclin 1 (protein) ↓ [55] Ileal LC3II (protein) ↓ [55] Ileal LC3II/LCRI ratio (protein) ↓ [55] Ileal p62 (protein) ↑ [55] Ileal autophagy signs (autophagosomes, autophagolysosomes, vacuoles) (transmission electron microscopy) ↓ [55]
HB-EGF	Intestinal TUNEL score (protein) ↓ [60] Intestinal CC3 score (protein) ↓ [60] Ileal cell migration (BrdU-positive cells) ↑ [59,61] Small intestinal integrin subunit α5 (mRNA) ↑ [61] Small intestinal integrin subunit β1 (mRNA) ↑ [61] Small intestinal integrin subunit α1 (mRNA) = [61] Small intestinal integrin subunit α2 (mRNA) = [61] Small intestinal integrin subunit α3 (mRNA) = [61] Small intestinal integrin subunit α6 (mRNA) = [61] Small intestinal integrin subunit α5 (protein) ↑ [61] Small intestinal integrin subunit β1 (protein) ↑ [61] Ileal epithelial cell proliferation (number of BrdU-positive cells) ↑ [59] Ileal Bax-to-Bcl-2 ratio (protein) = [37] Jejunal crypt epithelial cell proliferation (PCNA) ↑ [176] Jejunal number of enterocytes per villus ↑ [176] Jejunal number of LGR5+/prominin-1+ stem cells ↑ [176]
EPO	Ileal Beclin 1 immunoreactivity ↓ [194] Ileal LC3 immunoreactivity ↓ [194] Small intestinal Beclin 1 (protein) ↓ [194] Small intestinal LC3II (protein) ↓ [194] Ileal p62 immunoreactivity ↑ [194] Small intestinal p62 (protein) ↑ [194] Ileal CC3 immunoreactivity ↓ [194] Small intestinal Bcl-2 (protein) ↑ [194]
ATRA	Terminal ileal apoptosis intestinal crypts (TUNEL) ↓ [109] Terminal ileal proliferation crypt intestinal epithelial cells (Ki67, BrdU) ↑ [109]
Vitamin D	Intestinal cleaved caspase 3 (protein) ↓ [112] Intestinal Bcl-2 (protein) ↑ [112] Intestinal Ki67 (protein) ↑ [112]
**Probiotic interventions**	
*Bacteroides fragilis* strain ZY-312	Intestinal CC3 (protein) ↓ [113] Intestinal Bax (protein) ↓ [113] Intestinal Bcl-2 (protein) ↓ [113] Intestinal caspase 1 (protein) ↓ [113] Intestinal ASC (protein) ↓ [113] Intestinal NLRP3 (protein) ↓ [113] Intestinal IL1β (protein) ↓ [113] Intestinal gasdermin-D (protein) ↓ [113]
*Lactobacillus rhamnosus* GG	Ileal CC3 (protein) ↓ [161] Ileal apoptotic index (TUNEL) ↓ [161]
*Bifidobacterium bifidum* OLB6378	Ileal Bax (protein) ↓ [74] Ileal Bcl-w (protein) ↑ [74] Ileal Bax/Bcl-w ratio ↓ [74] Ileal CC3-positive cell number ↓ [74]
*Bifidobacterium breve* M-16V	Ileal caspase 3 (mRNA) ↓ [117]
**Other interventions**	
Amniotic fluid	Terminal ileal PCNA immunoreactivity ↑ [40]
Ginger	Intestinal TUNEL-positive cell number ↓ [121] Intestinal C3-positive cell number ↓ [121] Intestinal C8-positive cell number ↓ [121] Intestinal caspase 3 (protein) ↓ [121]
Fennel seed extracts	Terminal ileal C3-positive cells number ↓ [122] Terminal ileal C8-positive cells number ↓ [122] Terminal ileal C9-positive cells number ↓ [122] Intestinal C3 concentration (protein) ↓ [122]
Preterm human breast milk exosomes	Intestinal enterocyte proliferation (BrdU) ↑ [120]
Sesamol	Intestinal Bcl-2-positive cell number ↓ [125] Intestinal caspase-3 positive cell number ↓ [125]
Curcumin	Intestinal caspase 1 (protein) ↓ [124] Intestinal NLRP3 (protein) ↓ [124] Intestinal caspase 1 (mRNA) ↓ [124] Intestinal NLRP3 (mRNA) ↓ [124]

↑ depicts an increase, ↓ depicts a decrease; PUFA, polyunsaturated fatty acids; MPL, milk polar lipids; HMO, human milk oligosaccharides; EGF, epidermal growth factor; HB-EGF, hemoglobin-binding EGF-like growth factor; EPO, erythropoietin; ATRA, all-*trans* retinoic acid.

### 3.12. NEC Pathophysiology: Microbial Dysbiosis

Inappropriate microbial colonization or dysbiosis is considered to be an important factor contributing to NEC pathogenesis [1], although reports on the precise microbial colonization patterns or strains involved are conflicting [195]. A predominance of gram-negative bacteria from the phylum *Proteobacteria*, the class *Gammaproteobacteria* and the families *Enterobacteriaceae, Vibrionaceae* and *Pseudomonadaceae* are most strongly linked with NEC development [195]. Importantly, in a meta-analysis by Pammi et al. an increased relative abundance of the phylum *Proteobacteria* and a decrease of the phyla *Firmicutes* and *Bacteroides* were found prior to NEC onset [196]. In addition, a higher bacterial replication rate of all bacteria and especially *Enterobacteriaceae* has been linked to subsequent NEC development [197]. Although the intrauterine environment is not sterile [198], the major microbial colonization undoubtedly takes place in the first hours to days after birth and is influenced by various factors such as enteral feeding, gestational age, mode of delivery and antibiotic use [199,200]. The underdeveloped gut barrier of preterm born infants makes them vulnerable to the effects of a disturbed microbial colonization [195]. Mechanisms through which microbial dysbiosis can contribute to NEC pathogenesis include excessive TLR4 stimulation by endotoxin, disturbance of a balanced luminal short chain fatty acid (SCFA) content and changes in intestinal motility [201].

### 3.13. Enteral Feeding and Microbial Dysbiosis in Animal Models of NEC

Unfortunately, not many enteral feeding intervention studies have taken microbial changes into account (Table 11).

#### 3.13.1. Fat-Based Feeding Interventions

BCFA form a fat-based enteral feeding intervention known to influence microbial composition. Enteral treatment with BCFA increased the abundancy of *Bacillaeceae* and *Pseudomonadaceae* on family level and increased the relative abundance of *Bacillus subtilis* at species level in cecal samples of NEC protocol exposed rats to levels comparable to dam fed control animals [46]. In addition, relative abundancy of *Bacillus subtilis* was higher in healthy than in diseased animals [46]. Finally, BCFA administration increased the relative abundance of the species *Pseudomonas aeruginosa* to levels even higher than in dam fed animals [46]. As *Bacillus subtilis* is used as a probiotic, the BCFA induced increase in the relative abundance of this species is considered beneficial, for *Pseudomonas aeruginosa* this is unclear [46].

#### 3.13.2. Carbohydrate or Sugar-Based Feeding Interventions

As HMO are considered to be important prebiotics, it is not surprising these components have been studied in relation to intestinal microbial composition. Good et al. studied the effects of enteral treatment with the HMO 2′-FL in a murine NEC model on the abundancy of several microbial taxa in faecal content by 16S ribosomal RNA amplicon sequence analysis. They observed in NEC mice an increased abundancy of *Enterobacteriaceae* and decreased abundancy of *Lactobacillaceae* following HMO treatment [91]. However, the β-diversity was also reduced, indicating a more homogenous intestinal microbiome upon enteral HMO treatment [91]. In a pig NEC model, enteral administration of 2′-FL did not reduce cecal microbial colonization density and did not change microbial α-diversity in cecal tissue and cecal content, however, the proportion of genus *Enteroccocus* in cecal content was increased by administration of 2′-FL [133]. Also in a pig NEC model, administration of a mixture of >25 HMO components did not change the colonic relative abundance of different genera [202]. Administration of a mixture of four HMO did not change colonic microbial diversity (number of bacterial operational taxonomic units (OUT) per sample) [134] or the relative abundance of different genera [202] in a preterm pig model of NEC. Within the total microbial community no differences were observed in clustering, however, on the individual level HMO treated animals had a lower number of the genus *Fusobacterium* and this number was, although not statistically significantly, related to NEC development [134].

#### 3.13.3. Protein or Amino Acid-Based Feeding Interventions

In a preterm pig model of NEC, enriching formula feeding with caseinoglycomacropeptide (CGMP) or osteopontin (OPN) did not influence colon microbiota composition (similar α diversity and no significant changes in abundance of genera) [129].

#### 3.13.4. Hormone, Growth Factor or Vitamin-Based Feeding Interventions

Enteral vitamin A treatment in a murine NEC model had a strong influence on the microbial composition of intestinal tract content, accounting for 67.8% and 66.1% for the total variations observed on phylum and genus level, respectively [111]. Vitamin A treatment specifically decreased the abundance of the phylum *Proteobacteria* and the genera *Escherichia-Shigella*, *Lactobacillus*, *Acinetobacter* and *Gemella* and increased the phylum *Bacteroidetes* and the genera *Romboutsia*, *Bacteroides* and *Parabacteroides* [111] compared to control animals. The proportion of the phylum *Firmicutes* was not affected by vitamin A administration [111].

#### 3.13.5. Probiotic Feeding Interventions

In a quail NEC model, oral inoculation with *Bifidobacterium infantis-longum* decreased cecal bacterial counts of *Clostridium perfringens,* without altering counts of *Clostridium difficile* [130]. Administration of *Bacteroides fragilis* strain ZY-312 in a rat NEC model partially rescued the number of OTU in fecal samples, partially prevented a NEC induced reduction of the abundance of phylum *Bacteroidetes* and decreased the relative abundance of the phylum *Proteobacteria* [113]. In a rat NEC model, administration of *Lactobacillus reuteri* biofilms on maltose loaded microspheres shifted the fecal microbiome of NEC stressed rat pups more towards that of breastfed control pups than unbound *Lactobacillus reuteri* (16S rRNA sequencing analysis) [72]. On taxa level, *Lactobacillus* spp. abundance, which was negatively correlated to histological NEC severity, increased after *Lactobacillus reuteri* administration (both unbound and biofilm associated) and was more effectively maintained by administration of *Lactobacillus reuteri* as a biofilm on maltose loaded microspheres than by unbound *Lactobacillus reuteri* [72]. *Lactobacillus reuteri* bound to maltose loaded biospheres and unbound *Lactobacillus reuteri* effectively reduced the relative abundance of the potentially pathogenic *Enterobacter* spp. [72]. Finally, enteral administration of a mixture of probiotics (containing *Bifidobacterium animalis* and several *Lactobacillus species*) changed the general colonization pattern in distal ileum and colon (T-RFLP analysis), with a decrease in colonization density of *Clostridium perfringens*, and altered the relative proportion of several culturable bacteria [77]. It decreased the abundance of *Clostridia *(distal small intestinal homogenate and colon content) and *Enterococci* (stomach content and distal small intestinal homogenate) and increased the abundance of lactic acid bacteria (stomach content and colon content), *Lactobacilli* (stomach content and distal small intestinal homogenate) and total anaerobes (colon content) [77].

#### 3.13.6. Other Enteral Feeding Interventions

Enteral administration of amniotic fluid reduces distal small intestinal bacterial colonization in a pig model of NEC. In addition, colonic bacterial composition was changed towards controls by enteral administration of amniotic fluid [123].

**Table 11 nutrients-13-01726-t011:** Effect of enteral feeding interventions that affect microbial dysbiosis in experimental animal models of NEC.

Enteral Feeding Intervention	Effect on Microbial Dysbiosis (Compared to NEC Protocol Exposure without Feeding Intervention)
**Fat-based interventions**	
BCFA	Cecal *Bacillaeceae* (family) abundance ↑ [46] Cecal *Pseusomonadaceae* (family) abundance ↑ [46] Cecal *Bacillus subtilis* (species) abundance ↑ [46] Cecal *Pseudomonas aeruginosa* (species) abundance ↑ [46]
**Carbohydrate/sugar-based interventions**	
2′-FL	Fecal *Enterobacteriaceae* (family) abundance = [91] Fecal *Lactobacillaceae* (family) abundance = [91] Fecal microbiota β-diversity ↓ [91] Cecal microbial colonization density (FISH) = [133] α-Diversity cecal tissue = [133] α-Diversity cecal content = [133] Proportion *Enterococcus* (genus) in cecal content ↑ [133]
Mixture of four HMOs	Colonic microbial diversity (number of OTU per sample) = [134] Colonic microbial clustering = [134] Colonic relative abundance genera OTU = [202] Colonic number of *Fusobaceterium* (genus) on individual level ↓ [134]
Mixture of >25 HMO components	Colonic relative abundance genera OTU = [202]
**Protein/amino acid-based interventions**	
OPN	Colonic microbial α diversity = [129] Colonic microbial abundance of genera = [129]
CGMP	Colonic microbial α diversity = [129] Colonic microbial abundance of genera = [129]
**Vitamin-based interventions**	
Vitamin A	Fecal *Proteobacteria* (phylum) abundance ↓ [111] Fecal *Escherichia-Shigella* (genus) abundance ↓ [111] Fecal *Lactobacillus* (genus) abundance ↓ [111] Fecal *Acinetobacter* (genus) abundance ↓ [111] Fecal *Gemella* (genus) abundance ↓ [111] Fecal *Bacteroidetes* (phylum) abundance ↑ [111] Fecal *Bacteroides* (genus) abundance ↑ [111] Fecal *Romboutsia* (genus) abundance ↑ [111] Fecal *Parabacteroides* (genus) abundance ↑ [111]
**Probiotic interventions**	
*Bifidobacterium infantis-longum* strain CUETM 89-215	Cecal *Clostridium perfringens* (species) count ↓ [130] Cecal *Clostridium difficile* (species) count ↓ [130]
*Bacteroides fragilis* strain ZY-312	Fecal number of OTU ↑ [113] Fecal relative abundance *Bacteroidetes* (phylum) ↑ [113] Fecal relative abundance *Proteobacteria* (phylum) ↓ [113]
*Lactobacillus reuteri* DSM 20016	Fecal *Lactobacillus* (genus) abundance ↑ [72] Fecal *Enterobacter* (genus) abundance ↓ [72]
*Lactobacillus reuteri* biofilm on maltose loaded microspheres	Shift of fecal microbiome towards breastfed controls (16S sRNA sequencing) [72] Fecal *Lactobacillus* (genus) abundance ↑ [72] Fecal *Enterobacter* (genus) abundance ↓ [72]
Probiotic mixture (*Bifidobacterium animalis* DSM15954, *Lactobacillus acidophilus* DSM13241, *Lactobacillus casei* ATCC55544, *Lactobacillus pentosus* DSM14025 and *Lactobacillus plantarum* DSM13367	Distal small intestinal general colonization pattern (T-RFLP analysis) changed [77] Colonic general colonization pattern (T-RFLP analysis) changed [77] Distal small intestinal colonization density of *Clostridium perfringens* ↓ [77] Distal small intestinal homogenate relative proportion *Clostridium* (genus) ↓ [77] Colonic content relative proportion *Clostridium* (genus) ↓ [77] Distal small intestinal homogenate relative proportion *Enterococcus* (genus) ↓ [77] Stomach content relative proportion *Enterococcus* (genus) ↓ [77] Colon content relative proportion lactic acid bacteria ↑ [77] Stomach content relative proportion lactic acid bacteria ↑ [77] Distal small intestinal homogenate relative proportion *Lactobacillus* (genus) ↑ [77] Stomach content relative proportion *Lactobacillus* (genus) ↑ [77] Colon content relative proportion total anaerobes ↑ [77]
**Other interventions**	
Amniotic fluid	Distal small intestinal bacterial colonization (general eubacterial probe) ↓ [123] Colonic bacterial colonization normalized (PCA of T-RFLP analysis) [123]

↑ depicts an increase, ↓ depicts a decrease; BCFA, branched chain fatty acids; 2′-FL, 2′-fucosyllactose; HMO, human milk oligosaccharides; OPN, osteopontin; CGMP, caseinoglycomacropeptide.

### 3.14. NEC Pathophysiology: Disturbed Digestion and Absorption

Another factor that may contribute to NEC pathogenesis is carbohydrate maldigestion and malabsorption. Lactases and other disaccharidases are present at lower levels in premature infants than in term born infants, indicating that carbohydrate digestion in premature children is hampered [203]. In addition, there are some indications this may be further disturbed in infants that develop NEC. Book et al. found that infants with NEC have higher levels of fecal reducing substances, indicative of lactose malabsorption, than infants without gastrointestinal disease and higher levels were often detected before onset of clinical symptoms [204]. In tissue specimens from infants with NEC, no or only weak GLUT5, GLUT2 and lactase protein expression was observed, while these proteins were present in control tissue, whereas sucrose-isomaltase protein expression was preserved [167]. If carbohydrates such as lactose are not sufficiently digested and absorbed, they will reach the colon where they are subject to fermentation by colonic microbiota and lead to increased levels of fermentation products such as gasses (H_2_, CH_4_, CO_2_), SCFA and lactate [205]. These fermentation products could contribute to intestinal damage through local acidosis, stimulation of bacterial growth and potentially through induction of inflammation [206]. In line, a NEC study in preterm pigs showed that feeding with a maltodextrin-based formula that was malabsorbed, was associated with increased NEC incidence and severity, altered microbial and SCFA profiles compared to preterm pigs treated with a lactose-based formula that is easier to absorb [207]. Maldigestion and malabsorption can also result from NEC due to enterocyte loss or brush border destruction.

### 3.15. Enteral Feeding and Disturbed Digestion and Absorption in Animal Models of NEC

The influence of enteral nutritional interventions has been studied exclusively in pig models of NEC (Table 12). Importantly, in pigs and other large animals, changes in digestive enzyme activity and absorption in response to enteral nutrition seem to largely parallel that of human neonates [208,209], making large animal models particularly suitable for changes in digestion and absorption in the context of NEC.

#### 3.15.1. Carbohydrate or Sugar-Based Feeding Interventions

Currently, the only carbohydrate or sugar-based feeding intervention that has been shown to modestly influence digestion and absorption in experimental NEC are HMO. Enriching formula with a mixture of four HMO in a pig NEC model did not result in altered galactose or lactose absorption or brush border enzyme activities in the small intestine and did not change intestinal mRNA expression of sucrase, lactase, IAP and sodium/glucose transporter 1 (SGLT1) either [134]. However, colonic butyric acid concentrations slightly decreased after HMO administration [134]. A mixture of more than 25 HMO also did not change galactose and lactose absorption, but increased enzyme activity levels of lactase, aminopeptidase A, aminopeptidase N and dipeptidyl peptidase IV (DPPIV) in the distal small intestine compared to controls [134]. Feeding of preterm pigs with formula enriched by gangliosides or sialic acids (SL) did not rescue intestinal enzyme activity or intestinal hexose absorption in an experimental NEC model [100]. Finally, in a pig NEC model, enteral administration of 2′-FL did not improve galactose absorption or change the activity of several brush border enzymes [133].

#### 3.15.2. Protein or Amino Acid-Based Feeding Interventions

OPN, lactoferrin and CGMP were studied in relation to digestion and absorption in preterm pig models of NEC. No effects were seen of OPN enriched formula diet on digestive enzyme activity and intestinal hexose absorption [100,129]. A formula diet enriched with bovine lactoferrin neither changed intestinal absorption as measured by an oral bolus of galactose and lactose, nor changed brush border membrane enzyme activities in proximal, middle or distal small intestine [35]. In another pig NEC study, lactase activity in the middle part of the small intestine was increased by enteral supplementation of CGMP, while no effect was observed in the proximal or distal small intestine [129]. Plasma galactose levels upon an enteral bolus of galactose, suitable as a marker for hexose absorption, were increased in enteral CGMP supplementation, but this difference did not reach statistical significance [129].

#### 3.15.3. Probiotic Feeding Interventions

In a pig NEC model, enteral administration of a probiotic mixture containing Bifidobacterium animalis and several *Lactobacillus* strains increased distal intestinal enzyme activity of the brush border enzymes aminopeptidase A and aminopeptidase N without changing lactase and maltase enzyme activity [77].

#### 3.15.4. Other Enteral Feeding Interventions

Enteral feeding with formula supplemented with amniotic fluid increased maltase activity in the proximal and middle small intestine and increased galactose absorption compared to feeding with unsupplemented formula in a pig NEC model [123].

**Table 12 nutrients-13-01726-t012:** Effect of enteral feeding interventions that affect digestion and absorption in experimental animal models of NEC.

Enteral Feeding Intervention	Effect on Digestion and Absorption (Compared to NEC Protocol Exposure without Feeding Intervention)
**Carbohydrate/sugar-based interventions**	
Gangliosides	Intestinal enzyme activity small intestine = [100] Intestinal hexose absorption (galactose lactose absorption test) = [100]
SL	Intestinal enzyme activity small intestine = [100] Intestinal hexose absorption (galactose lactose absorption test) = [100]
HMO mixture of four components	Intestinal hexose absorption (galactose lactose absorption test) = [134] Intestinal enzyme activity small intestine = [134] Colonic butyric acid (protein) ↑ [134] Small intestinal sucrase (mRNA) = [134] Small intestinal lactase (mRNA) = [134] Small intestinal alkaline phosphatase (mRNA) = [134] small intestinal SGLT1 (mRNA) = [134]
HMO mixture >25 components	Intestinal hexose absorption (galactose lactose absorption test) = [134] Distal small intestinal lactase enzyme activity ↑ [134] Distal small intestinal aminopeptidase A enzyme activity ↑ [134] Distal small intestinal aminopeptidase N enzyme activity ↑ [134] Distal small intestinal dipeptidyl peptidase IV enzyme activity ↑ [134]
2′-FL	Galactose absorptive capacity = (galactose mannitol absorption test) [133] Proximal/middle/distal small intestinal enzyme activity (sucrose, maltase, lactase, ApN, ApA, DPPIV) = [133] Colon small intestinal enzyme activity (sucrose, maltase, lactase, ApN, ApA, DPPIV) = [133]
**Protein/amino acid-based interventions**	
OPN	Intestinal enzyme activity small intestine = [100,129] Intestinal hexose absorption (galactose lactose absorption test) = [100,129]
bovine lactoferrin	Small intestine enzyme activity = [35] Intestinal hexose absorption (galactose lactose absorption test) = [35]
CGMP	Middle small intestinal lactase enzyme activity ↑ [129] Proximal small intestinal lactase enzyme activity = [129] Distal small intestinal lactase enzyme activity = [129] Intestinal hexose absorption (galactose lactose absorption test) = [129]
**Probiotic interventions**	
Probiotic mixture (*Bifidobacterium animalis* DSM15954, *Lactobacillus acidophilus* DSM13241, *Lactobacillus casei* ATCC55544, *Lactobacillus pentosus* DSM14025 and *Lactobacillus plantarum* DSM13367)	Distal small intestinal lactase enzyme activity = [77] Distal small intestinal maltase enzyme activity = [77] Distal small intestinal ApA enzyme activity ↑ [77] Distal small intestinal ApN enzyme activity ↑ [77]
**Other interventions**	
Amniotic fluid	Proximal small intestine maltase enzyme activity ↑ [123] Middle small intestinal maltase enzyme activity ↑ [123]

↑ depicts an increase, ↓ depicts a decrease; SL, sialic acids; HMO, human milk oligosaccharides; 2′-FL, 2′-fucosyllactose; OPN, osteopontin; CGMP, caseinoglycomacropeptide.

### 3.16. NEC Pathophysiology: Enteric Nervous System Alterations

The enteric nervous system (ENS) is a large and complex division of the peripheral nerve system that resides in the gut [210]. It can morphologically be divided in the myenteric and submucosal plexus [210]. The ENS is involved in a variety of functions including gut motility, endocrine and exocrine secretions, microcirculation, regulation of immunity and gut barrier integrity [211,212,213]. Several studies have described alterations of the ENS in NEC. In intestinal segments of infants with NEC, morphological changes were observed in myenteric plexus, internal and external submucosal plexus concomitant with a loss of neurons and glial cells [214,215]. In addition, vasoactive intestinal peptide and NOS immunoreactivity was lost in the submucosal plexus of NEC patients [215]. A more recent study compared the myenteric plexus in tissue specimens from infants with NEC during acute disease and at the moment of stoma closure with gestational age matched control tissue [216]. Acute NEC was characterized by reduction of neuron and glial cell numbers per ganglion and a reduced number of nNOS expressing neurons [216]. Moreover, mRNA expression of nNOS and choline acetyltransferase (ChAT), two important regulators of intestinal motility, was reduced in acute NEC and increased CC3 immunoreactivity was present in both submucosal and myenteric plexus of acute NEC patients compared to control patients [216]. Although the total number of neurons per ganglion was recovered at the moment of stoma closure, this was not the case for the number of glial cells, the number of nNOS expressing neurons and nNOS mRNA expression [216]. Finally, Fagbemi et al. reports that ENS alterations in intestinal samples from infants with NEC are heterogeneous [217]. Whereas some infants had a disturbed architecture of the myenteric plexus with loss of mucosal and submucosal innervation and reduced expression of the glial cell marker glial fibrillary acidic protein (GFAP), no abnormalities were observed in samples from other affected children [217]. Although it is still unclear whether ENS alterations in NEC merely result from NEC or are involved in its pathophysiology, several findings support the latter scenario. First, ablation of glial cells is a plausible upstream target of NEC pathophysiology [218]. Second, in a rat model of NEC, neural stem cell transplantation reduced ENS alterations and was associated with improved intestinal motility and survival [216]. Third, in a preterm pig NEC model, region dependent changes in gut transit time were observed before radiological signs of NEC appeared, suggesting dysmotility may precede NEC development [219]. Enteroendocrine cells are chemo-sensing intestinal epithelial cells that play a key role in gastrointestinal secretion, motility and metabolism and signal amongst others through the ENS [220]. They are involved in the regulation of mucosal immunology and may be involved in NEC pathophysiology [221], although, in surgical NEC specimen, the number of enteroendocrine cells was not altered compared to controls [168].

### 3.17. Enteral Feeding and Enteric Nervous System Alterations in Animal Models of NEC

To date, only enteral feeding interventions with HB-EGF have been studied in relation to the ENS and enteroendocrine cells during NEC (Table 13). Enteral HB-EGF improved intestinal motility measured with a dye migration assay in a rat NEC model, although a reduction of total neuron counts in the ENS of NEC protocol exposed rats was not prevented by HB-EGF treatment [175]. In another study that used a rat NEC model, HB-EGF administration preserved the neuronal and glial cell integrity and nNOS expression and prevented neuronal degeneration and apoptosis during NEC [222]. Lastly, enteral administration of HB-EGF partially prevented NEC induced reduction of enteroendocrine cells in a rat model of NEC [176].

### 3.18. Enteral Feeding Interventions Affecting NEC Incidence and Severity in Human Studies

Many clinical trials have evaluated the effect of enteral nutritional interventions on NEC incidence or NEC related mortality (Table 14). Unfortunately, many interventions that are successful in animal models of NEC fail to show an effect in the clinical situation. Moreover, the certainty of evidence is often moderate to low and almost all studies are underpowered, which is likely to be, at least in part, responsible for the lack of successful enteral feeding interventions in clinical trials. Appendix A provides a detailed overview of the GRADE scoring of the evidence from clinical trials, the results are summarized in Table 14.

#### 3.18.1. Fat-Based Feeding Interventions

In a meta-analysis including 11 randomized controlled trials (RCTs) with *N* = 1753 neonates, supplementation of n-3 long chain PUFA did not result in a reduced NEC incidence [223]. The effect of n-3 long chain PUFA supplementation was more favourable in preterm infants ≤32 weeks, but did not reach statistical significance [223]. In a more recent large RCT, enteral supplementation with an emulsion rich in DHA also did not result in a reduced NEC incidence [224]. Certainty of evidence is low.

#### 3.18.2. Carbohydrate or Sugar-Based Feeding Interventions

In a meta-analysis, enteral administration of prebiotics (short-chain galacto-oligosaccharides (SC-GOS), long-chain fructo-oligosaccharides (LC-FOS), pectin-derived acidic oligosaccharides (pAOS), oligosaccharides, fructans, inulin or oligofructose) did not alter NEC incidence (RR 0.79 (95% CI 0.44–1.44)) [225] (low certainty of evidence).

#### 3.18.3. Protein or Amino Acid-Based Feeding Interventions

A recent meta-analysis including seven RCTs reported no difference in stage II or III NEC with enteral lactoferrin supplementation [226], however, with only a low grade of certainty (GRADE approach). A meta-analysis studying the effect of enteral and parenteral arginine administration on NEC incidence (3 RCTs included, two out of three RCTs exclusively studied enteral administration) observed a lower risk of NEC development with arginine treatment (relative risk (RR) 0.38, 95% CI 0.23–0.64, number needed to treat (NNT) 6) and a statistically significant reduction of death due to NEC (RR 0.18, 95% CI 0.03–1.00, NNT 20) [227], with a moderate/low certainty of evidence (GRADE approach). Enteral glutamine supplementation did not reduce NEC incidence in a meta-analysis [228]; certainty evidence was low (GRADE approach). Oral administration of IgG or a combination of IgG and IgA did not result in a reduced incidence of NEC (RR 0.84, 95% CI 0.57–1.25), need for NEC related surgery (RR 0.21, 95% CI 0.02–1.75) or death from NEC (RR 1.10, 95% CI 0.47–2.59) in a meta-analysis [229], with low to very low certainty of evidence (GRADE approach).

#### 3.18.4. Hormone, Growth Factor or Vitamin-Based Feeding Interventions

The effects of EPO were studied in a meta-analysis, in which no effect was found on NEC incidence (RR 0.62 (95% CI 0.15–2.59) [230]. Also in two more recent small RCTs, no effect was found of enteral EPO administration on NEC incidence [231,232]. A small RCT studying the effects of enteral granulocyte colony-stimulating factor (G-CSF) also did not find a reduced NEC incidence [233]. Another RCT did not find a reduction of NEC incidence with enteral supplementation of artificial amniotic fluid (rich in G-CSF) or artificial amniotic fluid and recombinant human EPO [231]. Lastly, two small RCTs studied the effects of oral supplementation of vitamin A with NEC incidence as a secondary outcome, but did not find differences ((RR 1.14, 95% CI 0.66–1.66) [234] and (RR 0.69, 95% CI 0.27–1.76) [235] respectively). For all these interventions, certainty of evidence was low or very low (GRADE approach).

#### 3.18.5. Probiotic Feeding Interventions

Probiotic enteral feeding interventions are increasingly used in the neonatal intensive care unit [236] and are the most studied group of enteral nutritional interventions for the reduction of NEC incidence [237]. In a recent systematic review and network meta-analysis including 56 RCTs (with in total *N* = 12,738 infants) reporting on severe NEC (stage II or higher), combinations of *Lactobacillus* spp. And *Bifidobacterium* spp. Or *Bifodobacterium animalis* subsp. *Lactis* were the most effective probiotic interventions [238]. Certainty of evidence was estimated to be moderate (GRADE approach). In addition, interventions using *Lactobacillus reuteri* or *Lactobacillus rhamnosus* were effective against severe NEC, although the effect size was lower than the aforementioned probiotic interventions [238] (moderate/low certainty of evidence). Interventions using a combination of *Lactobacillus* ssp., *Bifidobacterium* spp. and *Enterococcus* or a combination of *Bacillus* spp. and *Enterococcus* spp. reduced NEC incidence with the biggest effect size, however, with only low grade of certainty (GRADE approach) [238]. Another network meta-analysis observed statistically significant reduction of NEC incidence with probiotic interventions using *Bifidobacterium lactis* Bb-12 or B-94, *Lactobacillus reuteri* ATCC55730 or DSM17938, *Lactobacillus rhamnosus* GG, the combination of *Bifidobacterium bifidum, Bifidobacterium infantis, Bifidobacteirum longum and Lactobacillus acidophilus,* the combination of *Bifidobacterium infantis* Bb-02, *Bifidobacterium lactis* Bb-12 and *Streptococcus thermophilus* TH-4 and the combination of *Bifidobacterium longum* 35624 and *Lacobacillus rhamnosus* GG [239]. Certainty of evidence was estimated to be moderate to low (GRADE approach). In line with the evidence from this latter network meta-analysis, the European Society for Pediatric Gasteroenterology Hepatology and Nutrition (ESPGHAN) committee on nutrition and the ESPGHAN working group for probiotics and prebiotics at present conditionally recommend to provide either *Lactobacillus rhamnosus GG ATCC53103* or the combination of *Bifidobacterium infantis* Bb-02, *Bifidobacterium lactis* Bb-12, and *Streptococcus thermophilus* TH-4 as a preventive treatment to reduce NEC incidence [240].

#### 3.18.6. Other Feeding Interventions

In a relatively small multi-center RCT, enteral administration of carotenoids did not alter NEC incidence (OR 0.34 (95% CI 0.07–1.66) [241]. A mixture of probiotics, prebiotics and lactoferrin did reduce the overall NEC incidence and the incidence of NEC stage ≥2 in a small RCT ((RR 0.16 (95% CI 0.03–0.77) and RR 0.56 (95% CI 0.47–0.67) respectively) [242]. For both interventions, certainty of evidence was scored as low.

**Table 14 nutrients-13-01726-t014:** Enteral feeding interventions reducing NEC incidence or mortality in human studies.

Enteral Feeding Intervention		Effect on NEC Incidence			
Intervention	Study, *N*	Observed Relative Effect	Anticipated Absolute Effects		Certainty of Evidence (GRADE)
			Risk with No Enteral Feeding Intervention	Risk with Enteral Feeding Intervention (95% CI)	
Fat-Based Interventions					
n-3 PUFA	Meta-analysis, 11 RCTs, *N* = 1753 neonates	RR 1.17 (95% CI 0.77–1.79) (incidence all neonates) [223]	28 per 1000	5 more per 1000 more (from 6 less per 1000 to 50 more per 1000)	Low ⊕⊕⊝⊝ (risk of bias, imprecision)
		RR 0.50 (95% CI 0.23–1.10) (incidence neonates ≤32 weeks) [223]	58 per 1000	29 less per 1000 (from 45 less per 1000 to 6 more per 1000)	Low ⊕⊕⊝⊝ (risk of bias, imprecision)
DHA	RCT, *N* = 1205 neonates	RR1.16 (95% CI 0.79–1.69) (incidence) [224]	71 per 1000	11 more per 1000 (from 15 less per 1000 to 49 more per 1000)	Low ⊕⊕⊝⊝ (imprecision)
**Carbohydrate or sugar-based interventions**					
Prebiotics	Meta-analysis, 6 RCTs, *N* = 737 neonates	RR 0.79 (95% CI 0.44–1.44) (incidence) [225]	112 per 1000	24 less per 1000 (from 63 less per 1000 to 49 more per 1000)	Low ⊕⊕⊝⊝ (inconsistency, imprecision)
**Protein or amino acid-based interventions**					
Lactoferrin	Meta-analysis, 7 RCTs, *N* = 4874 neonates	RR 0.90 (95% CI 0.69–1.17) (incidence stage II or III NEC) [226]	47 per 1000	5 per 1000 less (from 15 per 1000 less to 8 per 1000 more)	Low ⊕⊕⊝⊝ (risk of bias, imprecision)
Arginine	Meta-analysis, 3 RCTs, *N* = 285 neonates	RR 0.38 (95% CI 0.23–0.64) (incidence) [227]	303 per 1000	188 per 1000 less (from 233 less per 1000 to 109 less per 1000)	Moderate ⊕⊕⊕⊝ (imprecision)
		RR 0.18 (95% CI 0.03–1.00) (death due to NEC) [227]	55 per 1000	45 less per 1000 (from 53 less per 1000 to 0 less per 1000	Low ⊕⊕⊝⊝ (imprecision)
Glutamine	Meta-analysis, 7 RCTs, *N* = 1172 neonates	RR 0.73 (95% CI 0.49–1.08) (incidence) [228]	90 per 1000	24 less per 1000 (from 46 less per 1000 to 8 more per 1000)	Low ⊕⊕⊝⊝ (imprecision, publication bias)
IgG or IgG + IgA	meta-analysis, 3 clinical trials, *N* = 2095 neonates	RR 0.84 (95% CI 0.57–1.25) (incidence) [229]	55 per 1000	9 less per 1000 (from 24 less per 1000 to 14 more per 1000)	Low ⊕⊕⊝⊝ (risk of bias)
		RR 0.21 (95% CI 0.02–1.75) (NEC related surgery) [229]	25 per 1000	20 less per 1000 (from 25 less per 1000 to 19 more per 1000)	Very low ⊕⊝⊝⊝ (risk of bias, imprecision)
		RR 1.10 (95% CI 0.47–2.59) (death due to NEC) [229]	10 per 1000	1 per 1000 more (from 5 less per 1000 to 16 more per 1000)	Very low ⊕⊝⊝⊝ (risk of bias, imprecision)
**Hormone/growth factor/vitamin-based interventions**					
EPO	Meta-analysis, 2 RCTs, *N* = 110 neonates	RR 0.62 (95% CI 0.15–2.59) (incidence stage II or III NEC) (NS) [230]	61 per 1000	24 less per 1000 (from 52 less per 1000 to 97 more per 1000)	Very low ⊕⊝⊝⊝ (inconsistency imprecision)
	RCT, *N* = 120 neonates	2.8pp increase (11.1% control group, 13.9% EPO group (NS) (incidence) [232]	111 per 1000	28 less per 1000 (no CI reported)	Very low ⊕⊝⊝⊝ (risk of bias, inconsistency imprecision)
	RCT, *N* = 100 neonates	2 pp reduction (8% control group, 6% EPO group) (NS) (incidence) [231]	80 per 1000	20 less per 1000 (no CI reported)	
EPO + G-CSF	RCT, *N* = 50 neonates	10 pp reduction (10% control group, 0% EPO group) (NS) (incidence) [233]	100 per 1000	100 per 1000 less (no CI reported)	Low ⊕⊕⊝⊝ (imprecision)
G-CSF	RCT, *N* = 50 neonates	10 pp reduction (10% control group, 0% EPO group) (NS) (incidence) [233]	100 per 1000	100 per 1000 less (no CI reported)	Low ⊕⊕⊝⊝ (imprecision)
G-CSF (in artificial amniotic fluid)	RCT, *N* = 100 neonates	2 pp reduction (8% control group, 6% EPO group) (NS) (incidence) [231]	80 per 1000	20 per 1000 less (no CI reported)	Very low ⊕⊝⊝⊝ (risk of bias, imprecision)
Vitamin A	RCT, *N* = 154 neonates	RR 1.14 (95% CI 0.66–1.66) (incidence) [234]	91 per 1000	13 per 1000 more (from 31 per 1000 less to 60 per 1000 more)	Low ⊕⊕⊝⊝ (risk of bias, imprecision)
	RCT, *N* = 262 neonates	RR 0.69 (95% CI 0.27–1.76) (incidence) [235]	77 per 1000	24 per 1000 less (from 56 per 1000 less to 59 per 1000 more)	
**Probiotic interventions**					
Probiotics	(network) meta-analysis, 56 RCTs, *N* = 12,738 neonates				
	*Lactobacillus* spp. and *Bifidobacterium* spp. 11 RCTs, *N* = 1878 neonates	OR 0.35 (95% CI 0.20–0.59) (incidence stage II or III NEC) [238]	63 per 1000	41 per 1000 less (from 51 per 1000 less to 26 per 1000 less)	Moderate ⊕⊕⊕⊝ (indirectness)
	*Bifodobacterium animalis* subsp. *Lactis* 5 RCTs, 628 neonates	OR 0.31 (95% CI 0.13–0.74) (incidence stage II or III NEC) [238]	94 per 1000	44 per 1000 less (from 56 per 1000 less to 16 per 1000 less)	Moderate ⊕⊕⊕⊝ (imprecision)
	*Lactobacillus reuteri* 5 RCTs, *N* = 1388 neonates	OR 0.55 (95% CI 0.34–0.91) (incidence stage II or III NEC) [238]	71 per 1000	28 per 1000 less (from 42 per 1000 less to 5 per 1000 less)	Low ⊕⊕⊝⊝ (imprecision, indirectness)
	*Lactobacillus rhamnosus* 5 RCTs, *N* = 839 neonates	OR 0.44 (95% CI 0.21–0.90) (incidence stage II or III NEC) [238]	60 per 1000	35 per 1000 less (from 50 per 1000 less to 5 per 1000 less)	Moderate ⊕⊕⊕⊝ (imprecision)
	Combination of *Lactobacillus* ssp., *Bifidobacterium* spp. and *Enterococcus* ssp. 7 RCTs, *N* = 1950 neonates	OR 0.28 (95% CI 0.16–0.49) (incidence stage II or III NEC) [238]	63 per 1000	46 per 1000 less (from 54 per 1000 less to 32 per 1000 less)	Low ⊕⊕⊝⊝ (risk of bias)
	Combination of *Bacillus* spp. and *Enterococcus* spp. 1 RCT, *N* = 355 neonates	OR 0.23 (95% 0.08–0.63) (incidence stage II or III NEC) [238]	110 per 1000	49 per 1000 less (from 59 per 1000 less to 23 per 1000 less)	Moderate ⊕⊕⊕⊝ (imprecision)
	Network meta-analysis 43 RCTs, *N* = 10,651 neonates				
	*Bifidobacterium lactis* Bb-12 or B-94 5 RCTs, *N* = 828 neonates	RR 0.25 (95% CI 0.10–0.56) (incidence) [239]	100 per 10000	75 per 1000 less (from 90 per 1000 less to 44 per 1000 less	Moderate ⊕⊕⊕⊝ (imprecision)
	*Lactobacillus reuteri* ATCC55730 or DSM17938 4 RCTs, *N* = 1459 neonates	RR 0.43 (95% CI 0.16–0.98) (incidence) [239]	92 per 1000	52 per 1000 less (from 77 per 1000 less to 2 per 1000 less)	Low ⊕⊕⊝⊝ (imprecision, indirectness)
	*Lactobacillus rhamnosus* GG 6 RCTs, *N* = 1507 neonates	RR 0.24 (95% CI 0.064–0.67) (incidence) [239]	24 per 1000	18 per 1000 less (from 22 per 1000 less to 8 per 1000 less)	Low ⊕⊕⊝⊝ (imprecision, indirectness)
	Combination of *Bifidobacterium bifidum, Bifidobacterium infantis, Bifidobacteirum longum* and *Lactobacillus acidophilus* 2 RCTs, *N* = 247 neonates	RR 0.25 (95% CI 0.051–0.89) (incidence) [239]	129 per 1000	97 per 1000 less (from 122 per 1000 less to 14 per 1000 less)	Low ⊕⊕⊝⊝ (imprecision)
	Combination of *Bifidobacterium infantis* Bb-02, *Bifidobacterium lactis* Bb-12 and *Streptococcus thermophilus* TH-4 2 RCTs, *N* = 1244 neonates	RR 0.29 (95% CI 0.073–0.78) (incidence) [239]	54 per 1000	38 per 1000 less (from 50 per 1000 less to 12 per 1000 less)	Low ⊕⊕⊝⊝ (inconsistency, imprecision)
	*Bifidobacterium longum* 35624 and *Lacobacillus rhamnosus* GG 2 RCTs, *N* = 285 neonates	RR 0.18 (95% CI 0.020–0.89) (incidence) [239]	42 per 1000	34 per 1000 less (from 41 per 1000 less to 5 per 1000 less)	Low ⊕⊕⊝⊝ (imprecision)
**Other interventions**					
Carotenoids	RCT, *N* = 229 neonates	OR 0.34 (95% CI 0.07–1.66) (incidence stage II or III NEC) [241]	52 per 1000	34 per 1000 less (from 48 per 1000 less to 60 per 1000 more)	Low ⊕⊕⊝⊝ (imprecision)
Mixture of probiotics, prebiotics and lactoferrin	RCT, *N* = 208 neonates	RR 0.16 (95% CI 0.03–0.77) (incidence) [242]	106 per 1000	89 per 1000 less (from 103 per 1000 less to 24 per 1000 less)	Low ⊕⊕⊝⊝ (imprecision)
		RR 0.56 (95% CI 0.47–0.67) (incidence stage II or III NEC) [242]	38 per 1000	17 per 1000 less (from 20 per 1000 less to 13 per 1000 less)	Low ⊕⊕⊝⊝ (imprecision)

⊕⊕⊕⊕ depicts high certainty of evidence, ⊕⊕⊕⊝ depicts moderate certainty of evidence, ⊕⊕⊝⊝ depicts low certainty of evidence, ⊕⊝⊝⊝ depicts very low certainty of evidence; PUFA, polyunsaturated fatty acids; DHA, docosahexaenoic acid; SC-GOS, short chain galacto-oligosaccharides; LC-FOS, long chain fructo-oligosaccharides; EPO, erythropoietin; G-CSF, granulocyte colony-stimulating factor; OR, odds ratio; RR, relative risk; CI, confidence interval.

### 3.19. Enteral Feeding Interventions Affecting Pathophysiological Mechanisms of NEC in Human Studies

Evidence from human studies on enteral feeding interventions that positively influence potential pathophysiological mechanisms behind NEC are sparse as it is difficult to study these outcome measures in (preterm) infants (Table 15). Nevertheless, overlap between mechanisms found in animal studies and effects observed in humans indicate evidence from animal studies likely provide insights valuable to the human NEC situation.

#### 3.19.1. Carbohydrate or Sugar Based Feeding Interventions

In a small RCT with 10 prebiotic supplemented and 13 only formula fed infants, 30 days of prebiotic supplementation of formula feeding with a mixture of SC-GOS and LC-FOS increased the percentage of gastric slow wave propagation measured with electrogastrography and decreased the gastric half emptying times inducing a gastrointestinal motility pattern comparable to breastmilk fed infants [243]. In another small RCT, enrichment of formula feeding with GOS and FOS decreased intestinal transit time (assessed by gastrointestinal passage of carmine red) [244]. In addition, stool viscosity was increased and stool pH was reduced, suggesting increased SCFA production by colonic fermentation upon GOS and FOS administration [244]. Enteral supplementation of SC-GOS, LC-FOS and acidic oligosaccharides (AOS) to preterm infants did not change fecal IL8 or calprotectin concentrations over time in a RCT [245].

#### 3.19.2. Protein or Amino Acid-Based Feeding Interventions

In a randomized controlled trial, infants orally treated with lactoferrin had a bigger increase in CD4+ CD25^high^ Foxp3+ Treg cells at discharge compared to controls [246]. In a double-blinded placebo-controlled trial, the effect of enteral administration of l-glutamine on intestinal barrier function was assessed with a dual sugar (mannitol, lactulose) absorption test. Both the urine recovery of lactulose and the ratio between urine recovery of lactulose and mannitol was lower after 7 and 30 days in infants treated with l-glutamine compared to placebo treated infants, demonstrating that L-glutamine positively influenced gut barrier function [247].

#### 3.19.3. Probiotic Feeding Interventions

Thirty days of prebiotic supplementation of formula feeding with a *Lactobacillus reuteri* normalized gastrointestinal motility by increasing the percentage of gastric slow wave propagation measured and decreasing the gastric half emptying in a small RCT with preterm infants [243]. In addition, in a small RCT, enteral administration of a formula with added *Bifidobacter lactis* improved intestinal barrier function (decreased lactulose mannitol ratio in urine) at 30 days postnatally [248].

#### 3.19.4. Other Feeding Interventions

Enteral administration of a mixture of probiotics, prebiotics and lactoferrin slightly increased systemic IFNϒ protein levels at 28 days of life, but did not affect several other cytokines (IL5, IL10 and IL17) in a RCT in premature infants [249].

**Table 15 nutrients-13-01726-t015:** Effect of enteral feeding interventions that affect pathophysiological mechanisms of NEC in human studies.

Enteral Feeding Intervention	Effect on Pathophysiological Mechanism (Compared to Placebo/No Intervention)
**Carbohydrate or sugar-based interventions**	
(SC) GOS + (LC) FOS	% Gastric slow wave propagation (electrogastrography) ↑ [243] Gastric half emptying time (echography) ↓ [243] Intestinal transit time (passage carmine red) ↑ [244] Stool pH ↓ [244] Stool viscosity ↑ [244]
SC GOS + LC FOS + AOS	Fecal IL8 = [245] Fecal calprotectin = [245]
**Protein or amino acid-based interventions**	
Lactoferrin	Whole blood CD4+ CD25high Foxp3+ Treg cell number ↑ [246]
L-Glutamine	Lactulose recovery in urine day 7 ↓ [247] Lactulose recovery in urine day 30 ↓ [247] Lactulose/mannitol recovery ratio in urine day 7 ↓ [247] Lactulose/mannitol recovery ratio in urine day 30 ↓ [247]
**Probiotic interventions**	
*Lactobacillus reuteri*	% Gastric slow wave propagation (electrogastrography) ↑ [243] Gastric half emptying time (echography) ↓ [243]
*Bifidobacterium lactis*	Lactulose/mannitol recovery ratio in urine day 30 ↓ [248]
**Other interventions**	
Mixture of probiotics, prebiotics and lactoferrin	Serum IL5 (protein) at 0, 14 and 28 days = [249] Serum IL10 (protein) at 0, 14 and 28 days (protein) = [249] Serum IL17 (protein) at 0, 14 and 28 days = [249] Serum IFNϒ (protein) 0, 14 days = [249] Serum IFNϒ (protein) 28 days ↑ [249]

↑ depicts an increase, ↓ depicts a decrease; SC-GOS, short chain galacto-oligosaccharides; LC-FOS, long chain fructo-oligosaccharides.

### 3.20. Interaction between Feeding Components and NEC

Despite the complex and rich composition of breastmilk and the concomitant presence of many bioactive factors [19,250] that are considered a major factor in the prevention of NEC, remarkably few studies have investigated the interaction or potential synergistic effect between two or more of these bioactive substances. In a quail NEC model, addition of FOS to the feeding enhanced the reduction of cecal *Clostridium perfringens* counts by *Bififobacteria,* an effect that was not observed by FOS administration in absence of *Bifidobacteria* [128,130]. In a recent meta-analysis, enteral supplementation of lactoferrin did not reduce the incidence of NEC (RR 0.90, 95% CI 0.69–1.17), whereas concomitant administration of lactoferrin and probiotics did result in a statistically significant reduction of NEC incidence (RR 0.04, 95% CI 0.00–0.62); however, these results need to interpreted with caution due to (very) low certainty of evidence [226,251]. Dvorak et al. investigated potential synergistic effects of EGF and HB-EGF in a rat model of NEC, but did not find additional protective effects against NEC [37]. Similarly, D’Souza et al. did not find benefits of combining enteral administration of the probiotic *Saccharomyces Boulardii* and GOS/FOS [135]. That combined enteral administration of nutritional components can also reduce the therapeutic effect was observed in a rat NEC model, where nucleotide administration abolished the PUFA induced reduction of mortality, gut necrosis, endotoxemia and intestinal PLA_2_ and PAFR mRNA expression [45].

### 3.21. Enteral Feeding Strategies and NEC: Feeding Regimens, Fortifiers and More

Besides the content of enteral nutrition, various other aspects of enteral feeding are likely to be related to the risk of NEC development and should be taken into account when designing trials studying enteral nutritional interventions for the prevention of NEC. Although evidence is not conclusive, factors that could be of relevance, especially for high risk populations such as ELBW infants, include the dose, duration and timing of trophic feeding/minimal enteral nutrition, the use of human milk-derived fortifiers, feed osmolality and standardized feeding regimens [252,253].

## 4. Discussion

Experiments in animal models of NEC provide a large amount of evidence of the beneficial effect of enteral nutritional interventions for preventing NEC incidence, severity, signs and symptoms, and mortality, as well as for ameliorating several pathophysiological processes related to NEC development including intestinal inflammation and intestinal barrier loss. A broad range of nutritional substances has been reported to be effective in several complementary experimental models, e.g., in different species and with different ways of inducing NEC. Especially HMO and growth factor-based interventions such as HB-EGF and EGF are promising as they have been shown to be effective in many experimental studies in which they target a broad range of pathophysiological mechanisms. Although some studies provide excellent insight in the underlying working mechanisms, addressing this for a broader range of interventions could be of great benefit to predict potential synergistic action between different substances of interest. This should therefore be subject of further research.

Despite the large amount of evidence from animal models, remarkably few enteral feeding interventions (e.g., arginine and probiotics) have been shown to be effective in meta-analyses of clinical trials. To date, only probiotics have reduced NEC incidence in adequately powered clinical studies and these interventions thereby form a promising preventive therapy, although even for these interventions certainty of evidence is at best moderate. Hence, the translation from preclinical findings in animal models to clinical practice remains challenging. Several underlying problems may be responsible for this arduous translation.

First, animal experiment related factors are in play. The current evidence from animal studies needs to be interpreted with caution, primarily due to the difficulty to adequately assess risk of bias in most animal studies and to determine certainty of evidence. Dissemination bias is likely present in animal studies of NEC, as researchers estimate that, in general, only around 50–60% of conducted animal studies [254,255] and data of only 26% of animals used are published [255]. Importantly, one of the main reasons for not publishing a study appears to be non-statistically significant results [254]. Moreover, other sources of bias may be present in experimental animal studies and are difficult to detect as many methodological aspects of the studies that are important for assessment of bias are poorly reported, both in studies incorporated in this systematic review and animal experiments in general [256]. Additionally, adequately assessing certainty of evidence from animal studies [33] is currently hampered, since amongst others confidence intervals and power calculations are often not reported. Due to (dissemination) bias, reports in literature of successful enteral feeding interventions in animal models may not reflect the true biological potential of the tested substance. Thus, based on the current evidence, it difficult to establish which preclinically studied interventions are most promising (considered safe, clinically relevant effect size, moderate to high certainty of evidence) and, hence, should be pursued in clinical trials. Besides, a smooth transition from animal research to clinical practice is hampered by the fact that experimental NEC modeling is still suboptimal. Notwithstanding the fact that many disease characteristics and a number of pathophysiological mechanisms involved in NEC are included in the current animal models of NEC, it is likely at least part of its complex pathophysiology is not adequately covered by the current models [257]. In addition, animal models are inherently limited due to the difficulty of using animals that are preterm and have bacterial colonization of the gut comparable to the human situation and differences between human and animal physiology [257,258,259,260].

Second, factors related to the conduct of clinical trials are involved. Many clinical trials are not designed with NEC incidence as primary outcome and are underpowered to convincingly prove a clinically significant beneficial effect. As Xiong et al. have nicely ascertained, the number of neonates required to prove a 20% relative risk reduction with 80% power assuming a 5% incidence of NEC is over 10,000 [237]. Including this amount of neonates in a study requires multi-center and international collaboration, which is logistically challenging and expensive. Moreover, NEC is not clearly defined and NEC diagnoses likely consist of a mixture of ‘classical’ NEC and closely related pathologies such as transfusion-related NEC, ischemic intestinal necrosis, spontaneous intestinal perforation and food protein intolerance enterocolitis syndrome [261,262]. It is likely that NEC(-like) diseases require a different treatment and that poorer effects of treatment will be found in clinical trials in which all these disease entities are pooled as one group.

Third, it is challenging to determine the optimal therapeutic regimen (dose, frequency, timing). Even though dose is of clear importance for the therapeutic effect [36,101], most animal studies only test a single dose and frequency of administration and it is therefore unclear how the dose and administration regimen used in animal studies should be translated to the human neonate. Of note, the optimal dose for the human neonate may be very well dependent on individual baseline levels, e.g., an infant with baseline deficit of a specific nutritional component may benefit from a higher dosage than an infant with baseline values within the normal range. Furthermore, timing of the feeding intervention often differs between animal studies and clinical trials. Due to the rapid nature of NEC progression following its onset, the value of nutritional interventions lies in prevention of NEC rather than treatment of ongoing NEC and as such, enteral feeding interventions are used as prevention in clinical trials. However, in animal models, enteral feeding interventions are almost always started in parallel to a NEC inducing protocol, and can therefore probably not be (fully) regarded as preventive. Studies looking at interventions at an earlier moment, such as in utero nutritional interventions, are in this context valuable [43].

Last, surprisingly few animal studies have looked at enteral feeding interventions with a combination of several bioactive substances, although this is, in light of the complex composition of breastmilk and the multifactorial nature of NEC pathogenesis, likely to be of pivotal importance.

Considering the abovementioned factors that hinder development of successful clinically applicable enteral nutritional interventions to reduce NEC incidence, several aspects should be improved. Future clinical trials investigating the potential of enteral feeding interventions to reduce NEC incidence should be adequately powered to at least be able to fairly estimate effect size and preferably reach statistical significance. In addition, clinical researchers should strive for the use of a clearer definition of NEC, ideally after international consensus regarding this definition in the field of NEC research. To this end, international collaboration between (pre)clinical NEC researchers and clinicians is essential.

Preclinical studies remain important to further understand NEC pathophysiology and optimize the current experimental models of NEC. In addition, the development of new human tissue based experimental models such as intestinal organoids, NEC-in-a-dish and gut-on-a-chip models is of importance [260,263,264]. In future preclinical experiments issues such as timing of intervention and dose/treatment regimen should be taken into account. Negative findings should be published, which could be stimulated by voluntary or mandatory registration of conducted (animal) studies as is more and more common practice in the clinical research field [255]. Moreover, the reporting quality of methodological aspects in experimental studies should be significantly improved to enable fair assessment of risk of bias and certainty of evidence. Finally, studying combinations of the most promising single substances based on findings in single component supplementation studies and on biological working mechanisms is likely to be of pivotal importance for finding effective enteral nutritional interventions that reduce clinical NEC incidence.

## Figures and Tables

**Figure 1 nutrients-13-01726-f001:**
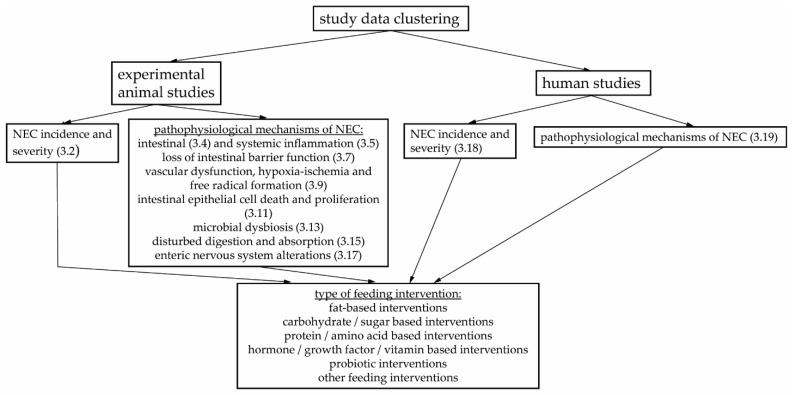
Overview of clustering of extracted data. The number in parentheses refers to the result section the data is incorporated in.

**Figure 2 nutrients-13-01726-f002:**
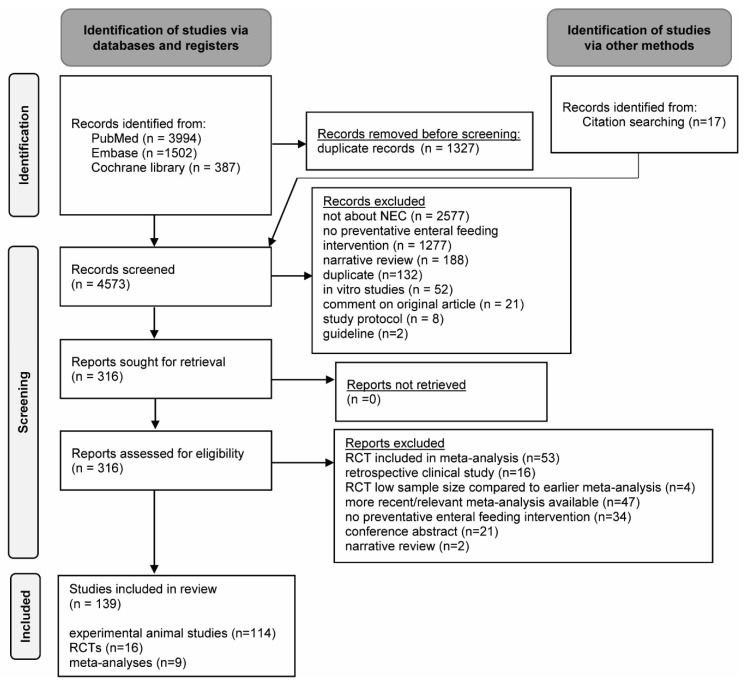
Flowchart of the article selection. Adapted from the PRISMA 2020 statement. Page, M.J.; McKenzie, J.E.; Bossuyt, P.M.; Boutron, I.; Hoffmann, T.C.; Mulrow, C.D.; Shamseer, L.; Tetzlaff, J.M.; Akl, E.A.; Brennan, S.E.; et al. The PRISMA 2020 statement: An updated guideline for reporting systematic reviews. *BMJ*
**2021**, *372*, n71 [34].

**Table 1 nutrients-13-01726-t001:** Enteral feeding interventions reducing NEC incidence in experimental animal models of NEC.

Fat-based interventions	AA and DHA [44] Egg phospholipids [44] PUFA [45] BCFA [46] Pomegranate seed oil [47] MFGM [48] DHA or EPA maternal intervention during pregnancy [43]
Carbohydrate/sugar-based interventions	HMO [49] GD3 [50]
Protein/amino acid-based interventions	Lactadherin [51] rPAF-AH [52]
Hormone/growth factor/vitamin-based interventions	EGF [37,41,53,54,55,56] HB-EGF [37,41,57,58,59,60,61,62] HGF [63] TGF-β1 [64] IGF1 [65] EPO [66]
Probiotic interventions	*Lactobacillus reuteri* DSM 17938 [67,68,69,70] *Lactobacillus reuteri* ATCC PTA 4659 [68] *Lactobacillus reuteri* biofilm on unloaded microspheres [71,72] *Lactobacillus reuteri* biofilm on MRS loaded microspheres [71] *Lactobacillus reuteri* biofilm on sucrose loaded microspheres [72] *Lactobacillus reuteri* biofilm on maltose loaded microspheres [72] *Bifidobacterium bifidum* OLB6378 [73,74] *Bifidobacterium infantis* [75] *Bifidobacterium adolescentis* [76] probiotic mixture (*Bifidobacterium animalis* DSM15954, *Lactobacillus acidophilus* DSM13241, *Lactobacillus casei* ATCC55544, *Lactobacillus pentosus* DSM14025 and *Lactobacillus plantarum* DSM13367) [77]
Other interventions	Amniotic fluid [63] Human breast milk extracellular vesicles [78] Berberine [79] Surfactant protein a [80] Human β-defensin-3 [81]

AA, arachidonic acid; DHA, docosahexaenoic acid; PUFA, polyunsaturated fatty acids; BCFA, branched chain fatty acids; MFGM, milk fat globule membrane; EPA, eicosapentaenoic acid; HMO, human milk oligosaccharides; GD3, ganglioside D3; rPAF-AH, recombinant platetet-activating factor acetylhydrolase; EGF, epidermal growth factor; HB-EGF, hemoglobin-binding EGF-like growth factor; HGF, hepatocyte growth factor; TGF-β1, transforming growth factor β1, IGF1, insulin-like growth factor 1.

**Table 2 nutrients-13-01726-t002:** Enteral feeding interventions improving histological injury scores in experimental animal models of NEC.

Fat-based interventions	Fish oil (rich in n-3 PUFA) [82,83] MPL [84] MFGM [48] Very low fat diet [85] Reduced long chain triacylglycerol diet (considered pre-digested) [85] Pomegranate seed oil [47]
Carbohydrate/sugar-based interventions	HMO [38,42,49,86,87,88,89,90] Neutral HMO (no sialic acids) [42] −2 HMO (two sialic acids) [42] DSLNT (HMO) [42] DSLNnT (synthetic disialyl glycan) [89,90] DS’LNnT (synthetic disialyl glycan) [89,90] 2′-FL [87,91,92] 6′-SL [92] 2′-FL and 6′-SL [42,92] Sialylated HMO [93] Sialylated GOS [87] GD3 [50] Hyaluronan 35 kD [94]
Protein/amino acid-based interventions	L-Glutamine/glutamine [84,95,96,97] Arginine [98,99] L-Carnitine [99] N-Acetylcysteine [85] Lactadherin [51] OPN [100] Lactoferrin [101] IAP [102,103]
Hormone/growth factor/vitamin-based interventions	EGF [53,54,56] Recombinant EGF from soybean extract [104] HB-EGF [41,58,59,60,62,105,106] HGF [63] relaxin [107] TGF-β1 [64] TGF-β2 [108] ATRA [109,110] Vitamin A [111] Vitamin D [112]
Probiotic interventions	*Bacteroides fragilis* ZY-312 [113] *Lactobacillus reuteri* DSM 17938 [68,69] *Lactobacillus reuteri* ATCC PTA 4659 [68] *Lactobacillus reuteri* biofilm on unloaded microspheres [71,72] *Lactobacillus reuteri* biofilm on MRS loaded microspheres [71] *Lactobacillus reuteri* biofilm on sucrose loaded microspheres [72] *Lactobacillus reuteri* biofilm on maltose loaded microspheres [72] *Bifidobacterium* microcapsules [114] *Bifidobacterium* mixture [115] *Bifidobacterium adolescentis* [76] *Bifidobacterium infantis* [116] *Bifidobacterium bifidum* OLB6378 [74] *Bifidobacterium breve* M-16V [117] *Lactobacillus rhamnosus* HN001 (live) [39] *Lactobacillus rhamnosus* HN001 (dead) [39] Lactobacillus rhamnosus isolated DNA [39] Probiotic mixture (*Bifidobacterium animalis* DSM15954, *Lactobacillus acidophilus* DSM13241, *Lactobacillus casei* ATCC55544, *Lactobacillus pentosus* DSM14025 and *Lactobacillus plantarum* DSM13367) [77] CpG-DNA [39]
Other interventions	Bovine milk exosomes [118] Native human breast milk exosomes [78,119] Pasteurized human breast milk exosomes [119] Preterm human breast milk exosomes [120] Ginger [121] Fennel seed extracts [122] Amniotic fluid [40,63,123] Curcumin [124] Sesamol [125] Astragaloside iv [126] Resveratrol [127] Berberine [79] Surfactant protein a [80] Human β-defensin-3 [81]

PUFA, polyunsaturated fatty acids; MPL milk polar lipids; MFGM, milk fat globule membrane; HMO, human milk oligosaccharides; DSLNT, disialyllacto-N-tetraose; 2′-FL, 2′-fucosyllactose; 6′-SL, 6′-sialyllactose; GOS: galacto-oligosaccharides; GD3, ganglioside D3; OPN, osteopontin; EGF, epidermal growth factor; HB-EGF, hemoglobin-binding EGF-like growth factor; HGF, hepatocyte growth factor; TGF-β1, transforming growth factor β1; TGF-β2, transforming growth factor β2; ATRA, all-*trans* retinoic acid.

**Table 3 nutrients-13-01726-t003:** Enteral feeding interventions reducing clinical disease score or signs and symptoms in experimental animal models of NEC.

Fat-based interventions	DHA and EPA [83] MFGM [48] Very low fat diet [85] Reduced long chain triacylglycerol diet (considered pre-digested) [85] MPL [84]
carbohydrate/sugar based interventions	2′-FL [91,92] 6′-SL [92] 2′-FL and 6′-SL [92] FOS [128] GD3 [50]
Protein/amino acid based interventions	Lactadherin [51] CGMP [129] OPN [129]
Hormone/growth factor/vitamin based interventions	EGF [54] HB-EGF [58] IGF1 [65] Vitamin D [112] Relaxin [107]
Probiotic interventions	*Lactobacillus reuteri* DSM17938 [70] *Bifidobacterium infantis-longum* strain CUETM 89-215 [130] *Bifidobacterium adolescentis* [76] *Bacteroides fragilis* ZY-312 [113] *Lactobacillus rhamnosus* HN001 (live) [39] *Lactobacillus rhamnosus* HN001 (dead) [39] *Lactobacillus rhamnosus* isolated DNA [39]
Other interventions	Ginger [121] Fennel seed extracts [122] Amniotic fluid [123] Sesamol [125] Human β-defensin-3 [81]

DHA, docosahexaenoic acid; EPA, eicosapentaenoic acid; MFGM, milk fat globule membrane; MPL milk polar lipids; 2′-FL, 2′-fucosyllactose; 6′-SL, 6′-sialyllactose; GD3, ganglioside D3; CGMP, caseinoglycomacropeptide; OPN, osteopontin; EGF, epidermal growth factor; HB-EGF, hemoglobin-binding EGF-like growth factor; IGF1, insulin-like growth factor 1.

**Table 4 nutrients-13-01726-t004:** Enteral feeding interventions improving survival in experimental animal models of NEC.

Fat-based interventions	PUFA [45] MFGM [48]
Carbohydrate/sugar-based interventions	HMO [42,49,88] Hyaluronan 35 kD [94]
Protein/amino acid-based interventions	Lactadherin [51] Lysozyme [131] rPAF-AH [52]
Hormone/growth factor/vitamin-based interventions	HB-EGF [41,58,59,62,105]
Probiotic interventions	*Bacteroides fragilis* ZY-312 [113] *Lactobacillus reuteri* DSM 17938 [68,69,132] *Lactobacillus reuteri* ATCC PTA 4659 [68] *Lactobacillus reuteri* biofilm on sucrose loaded microspheres [72] *Lactobacillus reuteri* biofilm on maltose loaded microspheres [72] *Bifidobacterium adolescentis* [76] *Bifidobacterium infantis* [75] *Bifidobacterium breve* M-16V [117]
Other interventions	Surfactant protein A [80] Human β-defensin-3 [81]

PUFA, polyunsaturated fatty acids; MFGM, milk fat globule membrane; HMO, human milk oligosaccharides; rPAF-AH, recombinant platelet-activating factor acetylhydrolase; HB-EGF, hemoglobin-binding EGF-like growth factor.

**Table 5 nutrients-13-01726-t005:** Overview of studies that did not report statistically significant preventative effects of enteral feeding interventions on NEC incidence, histological injury scores, clinical disease scores or signs and symptoms or survival in experimental animal models of NEC.

NEC Incidence	
Carbohydrate/sugar-based interventions	2′-FL [133] Gangliosides [100] SL [100] Lactose [92] Mixture of 4 HMO [134] Mixture of 25 HMO [134] IFOS [49]
Protein/amino acid-based interventions	OPN [100,129] CGMP [129] Bovine lactoferrin [35,36]
Probiotic interventions	*Lactobacillus reuteri* DSM 20016 [71,72]
Other interventions	amniotic fluid [123]
**NEC histological injury scores**	
Fat-based interventions	
Carbohydrate/sugar-based interventions	2′-FL [133] GOS [42,89] Lactose [38] 0 HMO (no sialic acids) [42] −1 HMO (one sialic acid) [42] −3 HMO (three sialic acids) [42] −4 HMO (four sialic acids) [42] Mixture of 4 HMO [134] Mixture of 25 HMO [134] 3‴-sLNnT [89] GD3 [89] DSLac [89] Neu5GC-DS’LNT [90] DS’LNnT [90] DSTa [90] DSGalB [90] Gangliosides [100] SL [100]
Protein/amino acid-based interventions	Bovine lactoferrin [35,36] (even higher score for [36]) OPN [129] CGMP [129]
Probiotic interventions	*Lactobacillus reuteri* DSM 20016 [71]
**NEC clinical disease score or signs and symptoms**	
Fat-based interventions	BCFA [46]
carbohydrate/sugar based interventions	Lactose [92] HMO [38,42] Mixture of four HMO [134] Mixture of 25 HMO [134] 2′-FL [133] GOS/FOS [135,136] GOS [42]
Protein/amino acid-based interventions	Glutamine [96] OPN [129] CGMP [129]
Probiotic interventions	*Saccharomyces Boulardii* [135,136]
Other interventions	Resveratrol [127]
**NEC survival**	
Fat-based interventions	Pomegranate seed oil [47] DHA [131] DHA or EPA maternal intervention during pregnancy [43]
Carbohydrate/sugar-based interventions	GOS/FOS [135] GOS [42] IFOS [49]
Hormone/growth factor/vitamin-based interventions	EGF [104,137]
Probiotic interventions	*Saccharomyces boulardii* [135] *Lactobacillus reuteri* DSM 20016 [72] *Lactobacillus reuteri* biofilm on unloaded microspheres [72]

DHA, docosahexaenoic acid; EPA, eicosapentaenoic acid; BCFA, branched chain fatty acids; HMO, human milk oligosaccharides; 2′-FL, 2′-fucosyllactose; SL, sialic acids; GOS, galacto-oligosaccharides; FOS, fructo-oligosaccharides; IFOS, infant formula oligosaccharides; GD3, ganglioside D3; EGF, epidermal growth factor; DSLac, disialyllactose; DSTa, disialyl T-antigen tetraose; 3‴-sLNnt, 3‴-sialyllacto-N-neotetraose; DSGalB, disialyl galactobiose; DS’LNnT, a2–6-linked disialyllacto-N-neotetraose; DS’LNT, a2–6-linked disialyllacto-N-tetraose; CGMP: caseinoglycomacropeptide; OPN: osteopontin.

**Table 7 nutrients-13-01726-t007:** Effect of enteral feeding interventions that reduce systemic inflammation in experimental animal models of NEC.

Enteral Feeding Intervention	Effect on Systemic Inflammation (Compared to NEC Protocol Exposure without Feeding Intervention)
**Carbohydrate/sugar-based interventions**	
HMO	Serum IL8 (protein) ↓ [49,88]
Hyaluronan 35 kD	Plasma TNFα (protein) ↓ [94] Serum CXCL1 (protein) ↓ [94] Serum IL12p70 (protein) ↓ [94] Serum IL6 (protein) ↓ [94] Serum IFNϒ (protein) ↓ [94]
**Protein/amino acid-based interventions**	
IAP	Serum TNFα (protein) ↓ (dose dependent) [162] Serum IL1β (protein) ↓ (dose dependent) [162] Serum IL6 (protein) ↓ (dose dependent) [162]
**Hormone/growth factor/Vitamin-based interventions**	
TGF-β	Serum IL6 (protein) ↓ [64] Serum IFNϒ (protein) ↓ [64]
**Probiotic interventions**	
*Bacteroides fragilis* ZY-312	Serum TNFα (protein) ↓ [113] Serum IFNϒ (protein) ↓ [113] Serum IL10 (protein) ↑ [113]
**Other interventions**	
Berberine	Serum IL6 (protein) ↓ [79] serum IL10 (protein) ↓ [79]
Human β-defensin-3	Serum TNFα (protein) ↓ [81]
Astragaloside IV	Serum TNFα (protein) ↓ [126] Serum IL6 (protein) ↓ [126] serum IL1β (protein) ↓ [126]

↑ depicts an increase, ↓ depicts a decrease; HMO, human milk oligosaccharides; IAP, intestinal alkaline phosphatase; TGF-β, transforming growth factor β.

**Table 13 nutrients-13-01726-t013:** Effect of enteral feeding interventions that affect the enteric nervous system in experimental animal models of NEC.

Enteral Feeding Intervention	Effect on Enteric Nervous System (Compared to NEC Protocol Exposure without Feeding Intervention)
**Hormone/growth factor/vitamin-based interventions**	
HB-EGF	Intestinal motility (dye migration assay) ↑ [175] Intestinal neuronal integrity Hu/D (protein) ↑ [222] Intestinal total neuronal count Hu/D = [175] Intestinal glial cell integrity GFAP (protein) ↑ [222] Intestinal nNOS expression (protein) ↑ [222] Intestinal neuronal apoptosis HuC/D TUNEL (protein) ↑ [222] Intestinal neuronal degeneration HuC/D FluoJade C (protein) ↑ [222] Jejunal entero-endocrine cells Chromogranin A (protein) ↑ [176]

↑ depicts an increase, ↓ depicts a decrease; HB-EGF, hemoglobin-binding EGF-like growth factor.

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
