# Peer review of "Enteral Feeding Interventions in the Prevention of Necrotizing Enterocolitis: A Systematic Review of Experimental and Clinical Studies"

_nutrients, 2021, doi:10.3390/nu13051726_

Round 1

Reviewer 1 Report

Authors have addressed most of the issues adequately. Few still require attention:

[1] Please add how many RCTs (and their sample sizes) were excluded because they did not met the following criteria as below and whether this has any influence on the findings, and conclusion:

"Selection criteria 2) the RCT was not included in a meta-analysis and was relatively 132 large (N = >50% of infants included in the meta-analysis)"

[2] Few issues in the section "Discussion:

  1. Discussion:  "Experiments in animal models of NEC provide overwhelming evidence of the beneficial effect of enteral nutritional interventions for preventing NEC.." Please add a line that the risk of bias and other flaws in the 'evidence' from animal studies. Given the limitations pointed out by the authors, the evidence from animal studies may not be reliable. The term/word 'Overwhelming' is best avoided
  • "Despite the convincing evidence from animal models, remarkably few enteral feeding interventions (e.g. arginine and probiotics) have been shown to be effective in meta-analysis of clinical trials.:.  If possible identify all enteral interventions in clinical practice that have 'adequate' or at least 'supporting evidence' from animal studies.
  • "Many clinical trials are not designed with NEC incidence as primary outcome and are underpowered to convincingly prove a clinically significant beneficial effect." It is therefore important to add this information in the tables summarising clinical and animal studies (se comment below)
  • "Third, not only is it difficult to establish which preclinically studied interventions are most promising and, hence, should be pursued in clinical trials". Consider adding  a line or two, to highlight what aspects of the animal studies will make it 'most promising' or ' ready for clinical studies in preterm infants
  • "Dissemination bias is likely present in animal studies of NEC, as researchers estimate that, in general, only around 50-60% of conducted animal studies are published (255, 256) and only 26%  the number of animals used (256). Please clarify the meaning of the last part of this line (i.e. only 26%... used")Add few lines on the most appropriate model/s for NEC in preterm infants?

It i s important to summarise evidence and its level/quality from clinical studies (RCTs and meta-analyses) in preterm infants. Use GRADE or any other standardised method. Add  a line whether such a system is available for providing 'level/quality of evidence' from animal studies.

Tables: For experimental as well as clinical studies, add sample size, primary outcome, and whether the study provided details of sample size estimation and power. (Use PICO format)

Language issues: Please check the manuscript for awkward, very long sentences and typos. (e.g. "Moreover, the reporting .....n experimental studies should be majorly improved to enable fair risk of bias assessment.". The word 'majorly' can be replaced by 'significantly')

Reviewer 2 Report

THank you for your attention to the recommendations to the paper.

Author Response

This manuscript is a resubmission of an earlier submission. The following is a list of the peer review reports and author responses from that submission.

Round 1

Reviewer 1 Report

I am having trouble with reference three and the statement that NEC is increasing due to increased survival of VLBW infants. The rates of NEC and Surgical NEC reported for all three hospital types and all records show a stable to decreasing rate of this serious disease.

enteral feeding is an important pharmacological target - I have never heard of enteral feeding management as a pharmacological intervention. It is a nutritional intervention with important physiologic and pathophysiologic consequences when manages in a variety of ways. - Or. do you mean enteral feedings in combination with pharmacological treatments (considering supplements as pharmaceuticals? Nutraceuticals?)

NEC onset is closely related to the initiation of enteral feeding - this could be interpreted as NEC starting closely after feedings are initiated as opposed to the likelihood of NEC developing is linked to the timing of initiation of enteral feedings - (Later onset of feedings increases risk of NEC)

In addition, supplementation of 191 formula with HB-EGF in a rat NEC model induced a dose-dependent reduction of NEC 192 incidence, with a therapeutic effect of moderate HB-EGF dosages that was not observed 193 with either a low or a high HB-EGF dose [48]. - This is an excellent example of the difficulty of nutritional research and how well designed interventional trials must be to be informative. Thank you for including it.

Intestinal inflammation is, in preclinical studies, the most extensively studied patho-279 physiological mechanism of NEC (Please add these two commas)

3.7.5. Probiotic feeding interventions 613

Many probiotic feeding interventions can improve intestinal barrier functions in the 614 context of NEC. In a rat NEC model, daily orogastric administration of Bifidobacterium 615 infantis reduced endotoxemia with a 10-fold at 48h. - I. think you mean reduced BY 10-fold.

3.21. Enteral feeding strategies and NEC: feeding regimens, fortifiers and more 1237

Besides the content of enteral nutrition, various other aspects of enteral feeding are 1238 likely to be related to the risk of NEC development and should be taken into account when 1239 designing trials studying enteral nutritional interventions for the prevention of NEC. Alt-1240 hough evidence is not conclusive, factors that could be of relevance, especially for high 1241 risk populations such as ELBW infants, include the dose and duration of trophic feed-1242 ing/minimal enteral nutrition, the use of human milk - I would consider changing to dose, duration and timing of trophic feed

IN the discussion when you are talking about dose dependent impact and the poor translation from animal to human studies I would also point out that understand the baseline for any given individual is likely important. Supplementing Omega three or vitamin A to a deficient population is much different and will lead to different impact than supplementing these to a sufficient population.

This paper represents a large amount of work and is very well organized and easy to follow. Congratulations to you for your accomplishment in this.

Reviewer 2 Report

Authors review of experimental and clinical studies of enteral feeding interventions for preventing NEC. I have the following comments. Hope they are helpful:

[1]  I have not heard of a 'semi-systematic' review. The reasons why this is not a conventional systematic review need to be acknowledged.

[2] Of the total 55 pages, 37 are devoted to animal /experimental studies, and only 4 to clinical studies. This is probably because they have included 'meta-analyses' wherever possible. The rationale for this could be clearer in methodology section.

[3] The last paragraph of 'Introduction' (Page 2) does not lead to a clear and justified 'Aim'. Ideally it should  clarify why the focus on the 13 points (From 'Incidence' and 'Histological injury' to issues with 'Digestion and absorption, and 'dysfunction of the enteric nervous system i.e. Tables 2 to 13) These 12 headings are better covered briefly under pathophysiology of NEC to point the opportunity for interventions [i.e. Fat, proteins, hormone-based, probiotic interventions, and 'Other'] in 'Animal/Experimental' vs. 'Clinical' studies'. This will help readers to grasp the manuscript. At present the big junks of dry text, long list of headings and subheadings make it difficult to understand the focus. A simple flow diagram will help in demonstrating the methodology

[4] The major scientific limitation of the manuscript is the lack of robust assessment of the animal/experimental studies, and of the 'systematic reviews' of clinical trials, and the included extra RCTs that were not covered in the meta-analysis. Authors could refer to the standardised methodology for critical review of Systematic reviews, of RCTs, and importantly of animal studies. The current descriptive approach seriously compromises the validity of the results despite the massive effort.

[5] The 'Results' section starts rather abruptly and the subheadings for 3.1 seem a bit out of place. This is probably because they relate to the pathophysiology  and subsequently proposed opportunities for 'interventions'. 'Dysbiosis' is missing here.

[6] Why IP melatonin with prostaglandin was included is not clear. Similarly why IV and intraperitoneal interventions were 'excluded' (because the focus was only on enteral feeding interventions) could be clarified. The  statements about RCTs with sample size >1000 or meta-analysis with n<500 are confusing.

[7] Table 1:t is not clear why the focus was only on the interventions included in the search strategy.  It is the lack of a pre-specified standardised approach that compromises the scientific value of the work.

[8] I appreciate the  massive work that has gone in presenting the results from animal/experimental studies (Tables 2 to 13] but without assessment of the quality of those studies [risk of bias etc) it is difficult to provide a reliable scientific 'summary' statement on level/quality of evidence from such studies.

[9] Authors quote pubmed and Cochrane as the databases they have searched , which is not comprehensive.  Cochrane database will not cover animal studies.

Overall, the manuscript needs significant work to improve validity of its results and hence conclusion. It may be better to focus only on animal/experimental studies